# The Interplay of Cardiometabolic Syndrome Phenotypes and Cardiovascular Risk Indices in Patients Diagnosed with Diabetes Mellitus

**DOI:** 10.3390/ijms26136227

**Published:** 2025-06-27

**Authors:** Daniela Denisa Mitroi Sakizlian, Lidia Boldeanu, Adina Mitrea, Diana Clenciu, Ionela Mihaela Vladu, Alina Elena Ciobanu Plasiciuc, Andra Veronica Șarla, Isabela Siloși, Mihail Virgil Boldeanu, Mohamed-Zakaria Assani, Daniela Ciobanu

**Affiliations:** 1Doctoral School, University of Medicine and Pharmacy of Craiova, 200349 Craiova, Romania; mitroidenisa96@gmail.com (D.D.M.S.); andrasarla98@gmail.com (A.V.Ș.); 2Department of Microbiology, Faculty of Medicine, University of Medicine and Pharmacy of Craiova, 200349 Craiova, Romania; lidia.boldeanu@umfcv.ro; 3Department of Diabetes, Nutrition and Metabolic Diseases, Faculty of Medicine, University of Medicine and Pharmacy of Craiova, 200349 Craiova, Romania; ada_mitrea@yahoo.com (A.M.); dianaclenciu@yahoo.com (D.C.); ionela.vladu@umfcv.ro (I.M.V.); 4Department of Health and Motricity, Faculty of Medical and Behavioral Sciences, “Constantin Brâncuși” University of Târgu-Jiu, 210185 Târgu-Jiu, Romania; 5Department of Immunology, Faculty of Medicine, University of Medicine and Pharmacy of Craiova, 200349 Craiova, Romania; isabela_silosi@yahoo.com (I.S.); mihail.boldeanu@umfcv.ro (M.V.B.); 6Department of Internal Medicine, University of Medicine and Pharmacy of Craiova, 200349 Craiova, Romania; elada192@yahoo.com

**Keywords:** metabolic syndrome, cardiometabolic risk, triglyceride-glucose index, atherogenic index of plasma, type 2 diabetes mellitus, prediabetes, cardiometabolic phenotypes, cardiovascular risk

## Abstract

Metabolic syndrome (MetS) and its associated cardiometabolic phenotypes significantly contribute to the global burden of cardiovascular disease (CVD), especially in individuals with type 2 diabetes mellitus (T2DM) and prediabetes. This study aimed to explore the association between cardiometabolic phenotypes—specifically, metabolically unhealthy normal weight (MUHNW) and metabolically unhealthy obese (MUHO)—and various cardiovascular risk indices including the triglyceride-glucose (TyG) index and its derivatives, the atherogenic index of plasma (AIP), the cardiometabolic index (CMI), and the cardiac risk ratio (CRR). A total of 300 participants were evaluated (100 with prediabetes and 200 with T2DM). Anthropometric, biochemical, and lifestyle parameters were assessed and stratified across phenotypes. The results demonstrated that cardiovascular risk indices were significantly elevated in the MUHO compared to MUHNW phenotypes, with T2DM patients consistently exhibiting higher risk profiles than their prediabetic counterparts. TyG-derived indices showed strong correlations with BMI, waist–hip ratio (WHR), waist–height ratio (WHtR), and body fat percentage (%BF). The findings suggest that cardiometabolic phenotypes are more strongly associated with elevated cardiometabolic risk indices than body weight alone. These indices may enhance early risk stratification and intervention efforts. The study investigates the association of cardiometabolic phenotypes with surrogate cardiovascular risk indices, not direct CVD outcomes, However, the cross-sectional design and population homogeneity limit the generalizability of the results and preclude causal inference.

## 1. Introduction

Metabolic syndrome (MetS) stands as one of the most significant challenges confronting public health systems globally, impacting over a billion adults across both developed and developing nations [1,2,3,4].

MetS encompasses a cluster of metabolic disorders, including central obesity, insulin resistance (IR), hypertension, and dyslipidemia. This constellation of risk factors significantly elevates the likelihood of cardiovascular morbidity and mortality, as well as all-cause mortality. Individuals with MetS are at increased risk of acute cardiovascular events, including myocardial infarction and stroke, in addition to a heightened predisposition to the development of type 2 diabetes mellitus (T2DM) [1,5,6,7,8].

The Atherogenic Index of Plasma (AIP) is a critical biomarker that provides insights into several pathological conditions. It has established correlations with risk of cardiovascular disease (CVD), chronic kidney disease, and metabolic dysregulation. Research indicates that elevated AIP levels are associated with a higher prevalence of hypertension, T2DM, and cognitive impairment. Therefore, regular monitoring of AIP could be pivotal for early detection and proactive management in populations at elevated risk [9].

The Cardiometabolic Index (CMI) is a groundbreaking metric for assessing obesity, integrating both visceral adiposity and lipid profiles into a comprehensive measure. This index is particularly relevant in understanding the interplay between factors influencing cardiac metabolism, which are critical in the pathophysiology of heart disease. By providing a robust framework for evaluating cardiac metabolic risk, the CMI serves as a valuable tool in both clinical and research settings [10]. The Cardiac Risk Ratio (CRR) is a key metric utilized in the assessment of CVD risk, providing a quantitative evaluation of an individual’s risk profile based on various clinical and demographic factors associated with cardiovascular health [11].

MetS, along with its comorbidities, such as CVD and non-alcoholic fatty liver disease (NAFLD), represents a considerable economic strain on healthcare systems. A simulation study conducted in Spain in 2025 projected that a 15% reduction in body weight among individuals with obesity could result in healthcare savings of approximately €105 million per 100,000 individuals over a decade. This financial benefit is largely attributable to decreased incidences of T2DM and hypertension. Since obesity is a pivotal factor in the development of MetS, these findings underscore the potential economic advantages of targeting MetS-related risk factors in public health interventions [12].

MetS engages intricate pathogenic pathways, with IR recognized as the primary mechanism. Key factors contributing to MetS include oxidative stress, chronic inflammation, endothelial dysfunction, and alterations in lipid metabolism. These interconnected mechanisms underscore the complex and multifaceted pathophysiology of MetS [13,14,15].

Cardiometabolic syndrome (CMS) represents a critical risk factor for CVD, substantially increasing the risk of both cardiovascular and overall mortality. This syndrome is defined by a cluster of metabolic abnormalities, including central obesity, hypertension, and dyslipidemia, characterized by elevated triglycerides, decreased high-density lipoprotein cholesterol (HDL-C), and glucose intolerance. Currently, only 6.8% of the adult population meets the criteria for optimal cardiometabolic health. Excessive adiposity, especially when centrally located, is recognized as a significant independent factor exacerbating various metabolic abnormalities linked to CVD. These disturbances include dyslipidemia, hypertension, hyperglycemia, IR, and systemic inflammation [16].

IR is characterized by a diminished sensitivity to insulin, resulting in impaired glucose uptake and the subsequent elevation of blood glucose levels. This condition is recognized as both a contributing factor and a prognostic indicator of adverse outcomes in patients with CVD, regardless of whether they have diabetes [17]. The global burden of CVD continues to escalate, with the incidence of cases increasing from 271 million in 1990 to 523 million by 2019. As a result, the deaths associated with CVD have increased from 12.1 million to 18.6 million during this period. This substantial rise in both illness and death rates highlights the urgent public health issue posed by CVD globally [18].

Managing cardiovascular risk in individuals with T2DM is becoming ever more important. Although there are several clinical risk scoring systems designed to forecast cardiovascular events, a standardized risk stratification tool tailored for T2DM patients is still lacking. Research has indicated that the coronary artery calcium score (CACS), assessed through coronary computed tomography, offers incremental prognostic value for cardiovascular events in this population, surpassing the predictions made by the widely utilized Framingham risk score (FRS). The most recent guidelines from the American Heart Association and American College of Cardiology (AHA/ACC) endorse the application of the CACS for risk assessment in intermediate-risk individuals when traditional risk stratification yields ambiguity. Accordingly, the CACS may serve as a pivotal biomarker for cardiovascular risk stratification in T2DM patients [19,20,21,22,23,24,25].

In the investigation of MetS, various phenotypic classifications have been established to account for the heterogeneity associated with body mass index (BMI). While BMI serves as a prevalent metric for evaluating CVD risk, it is important to recognize that individuals classified within identical BMI brackets may exhibit distinct cardiometabolic risk profiles. This underscores the need for more nuanced approaches to risk assessment beyond traditional BMI categorizations [26,27]. In response to the heterogeneity observed in metabolic health profiles, distinct classifications such as metabolically healthy normal weight (MHNW), metabolically unhealthy normal weight (MUHNW), metabolically healthy obese (MHO), and metabolically unhealthy obese (MUHO) have been established. These phenotypes emphasize the importance of metabolic health and the evaluation of cardiometabolic risk factors, moving beyond the traditional reliance on BMI as a sole indicator [28,29,30,31].

The phenotypes under consideration, which encompass metabolic and cardiovascular risk factors, establish a robust framework for our investigation. Our study focuses on elucidating the established association between the triglyceride-glucose index (TyG) and MetS, alongside key cardiovascular risk indices such as the CRR, CMI, and AIP. To the best of our knowledge, it is the first study to compare TyG, TyG-derived indices, AIP, CMI, and CRR, in the context of the different phenotypes of the cardiometabolic syndrome. Our objective was to analyze and compare cardiometabolic risk indices among different phenotypes of individuals with T2DM and those in a prediabetic state.

Previous research has established associations between these indices and CVD risk; however, our objective was to assess the variability of these metrics across specific phenotypes in populations with T2DM and prediabetes (PreDM).

## 2. Results

### 2.1. Comprehensive Analysis of PreDM and T2DM Patients Presenting with Metabolic Syndrome: Demographic and Clinical Characteristics, Medical History, and Laboratory Assessments

In the current study, we examined a cohort of 200 patients diagnosed with T2DM and MetS, with a mean age of 62.91 years and a standard deviation (SD) of 12.32, as detailed in Table 1.

The sample included a balanced distribution of 101 males and 99 females. The control group, referred to as the PreDM group, had a mean age of 56.51 years and a SD of 11.52, with males representing 47% of this population. A significant age difference was identified between the two groups (*p* < 0.0001); however, gender distribution did not show a significant difference (*p* = 0.56).

In terms of the patients’ residential distribution, it was observed that the majority of individuals in both the T2DM and PreDM cohorts resided in urban environments. Additionally, statistical analysis revealed no significant difference in residential status between the two groups, with a *p*-value of 0.75.

An individual’s medical history is a significant differentiator between the T2DM cohort and the PreDM cohort, supported by robust statistical analysis. Comorbidities such as hypertension, dyslipidemia, and hepatosteatosis were present in over two-thirds of the patients evaluated.

In relation to habitual behaviors, the statistical analysis revealed significant associations for both alcohol consumption and tobacco use, with *p*-values of 0.03 and 0.01, respectively.

The analysis of laboratory parameters indicated that the diagnostic thresholds for T2DM were significantly elevated compared to those of the PreDM cohort, as shown in Table 1 (*p* < 0.0001). Notably, inflammatory markers demonstrated statistical significance, with an erythrocyte sedimentation rate (ESR) of *p* = 0.02 and C-reactive protein (CRP) at *p* < 0.0001.

Furthermore, white blood cell (WBC) counts also achieved significance, yielding a *p*-value of 0.003. In the context of hepatosteatosis, there appears to be an associated alteration in liver enzyme levels, evidenced by an aspartate aminotransferase (AST) *p*-value of 0.003 and an alanine aminotransferase (ALT) *p*-value of 0.0004.

### 2.2. A Comparative Examination of Clinical Features, Anthropometric Measurements, Metabolic Indices, Lipid Profiles, and Metabolic Syndrome-Related Parameters in Individuals with PreDM and T2DM

Both cohorts adhered to a stringent medication regimen, resulting in no significant differences observed between them in terms of systolic blood pressure (SBP) and diastolic blood pressure (DBP) during the examination. In the analysis of anthropometric parameters, including height, weight, waist circumference (WC), and hip circumference (HC), along with obesity-related indices such as waist-to-height ratio (WHtR), waist-to-hip ratio (WHR), and BMI, we observed statistically significant differences between the two groups. Notably, the mean values for WHtR showed significance (*p* = 0.02), while WHR approached significance with a *p*-value of 0.052.

During our investigation, we performed an extensive analysis of lipemic profiles, revealing significant variations in lipid levels between the two cohorts: individuals diagnosed with T2DM and those classified as having PreDM. The total cholesterol (TC) levels exhibited a statistically significant difference (*p* = 0.001), as did low-density lipoprotein cholesterol (LDL-c) (*p* = 0.005) and high-density lipoprotein cholesterol (HDL-c) (*p* < 0.0001), as it is shown in Table 2.

Although the total triglyceride (TG) levels did not present a statistically significant difference, those with T2DM had higher values compared to the PreDM group, with median TG levels of 128.50 mg/dL (range: 45.00–610.00) versus 144.50 mg/dL (range: 44.00–1563.00) in the PreDM cohort.

When evaluating the MetS-related indices, all metrics demonstrated statistical significance except for TyG adjusted for WC (TyG-WC) and TyG adjusted for WHtR (TyG-WHtR). Specifically, TyG showed a highly significant *p*-value (<0.0001), followed by TyG adjusted for BMI (TyG-BMI) (*p* = 0.02), TG/HDL-c ratio (*p* = 0.0005), AIP (*p* = 0.0005), CMI (*p* = 0.008), and CRR (*p* = 0.02).

### 2.3. Comparative Analysis of the Two Phenotypes of Cardiometabolic Syndrome: MUHNW and MUHO in Individuals with PreDM and T2DM

The two groups were further stratified into two subgroups according to their cardiometabolic phenotypes: MUHNW and MUHO, as detailed in the accompanying Table 3.

The PreDM cohort exhibited a statistically significant difference in sex distribution, with a *p*-value of 0.05. Additionally, the presence of hepatosteatosis also demonstrated a significant difference at the same *p*-value of 0.05.

In contrast, within the T2DM subgroup, several variables approached statistical significance: age (*p* = 0.057) and alcohol consumption (*p* = 0.054). Moreover, significant differences were noted for hepatosteatosis, with a *p*-value of less than 0.0001, and educational background (*p* = 0.03).

Regarding anthropometric measurements and indices, while the T2DM cohort showed significant differences across all metrics, the PreDM cohort exhibited significant variations for height, which showed a *p*-value of <0.0001 compared to 0.001 for T2DM; weight registered for both as *p* < 0.0001; and for T2DM, WC, HC, and WHR, all *p* < 0.0001.

Furthermore, WHtR had *p*-values of 0.001 versus <0.0001, while BMI and %BF both had *p*-values of <0.0001 for both groups. There were no statistically significant differences in laboratory examination for the PreDM cohort, while for the T2DM, a significant difference was noted in the values of CRP (*p* = 0.01), AST (*p* = 0.05), ALT (*p* < 0.0001), eGFR (*p* = 0.05), and HGB (*p* = 0.009).

In the PreDM cohort, we found significant statistical differences solely in TyG-WHtR, with 0.0003. In contrast, among individuals with T2DM, all MetS-related metrics were markedly elevated in those with the MUHO phenotype.

Specifically, we observed significant differences in the TG/HDL-c ratio and AIP (*p* = 0.001), along with the CRR (*p* = 0.04). Additionally, indices such as the TyG-WHtR and CMI exhibited highly significant variations (*p* < 0.0001).

### 2.4. Comparative Analysis of PreDM MUHNW Patients with T2DM MUHNW Patients and PreDM MUHO Patients with T2DM MUHO Patients

In the comparative analysis of MUHNW phenotypes, age emerged as a statistically significant variable with a *p*-value of 0.0007, alongside the presence of hepatosteatosis (*p* = 0.01).

Conversely, for MUHO phenotypes, both age and the prevalence of hepatosteatosis showed notable statistical differences, with *p*-values of 0.005 and 0.0005, respectively. Additionally, smoking and alcohol consumption exhibited significant distinctions in MUHO phenotypes, with *p*-values of 0.01 and 0.007, respectively.

Furthermore, when analyzing MUHWN phenotypes, the PreDM subgroup demonstrated significant increases in height and WHtR, with *p*-values of 0.04 and <0.0001, respectively.

Within the MUHNW phenotypes, significant differences were noted in the TyG-associated index: TyG-WHtR (*p* < 0.0001). In contrast, for MUHO phenotypes, the next indices achieved statistical significance: TG/HDL-c ratio (*p* = 0.0002), AIP (*p* = 0.001), CMI (*p* = 0.0008), and CRR (*p* = 0.01) (refer to Figure 1 and Table 4).

### 2.5. Connections of TyG and TyG-Related Indices with BMI, WHR, WHtR, and Body Fat Percentage in the PreDM and T2DM Groups

The accompanying Table 5 and Table 6 illustrate the intricate relationships between the TyG, its related indices (TyG-BMI, TyG-WC, and TyG-WHtR), and various anthropometric measures, including BMI, WHR, WHtR, and body fat percentage (%BF), within cohorts of individuals diagnosed with PreDM and T2DM.

BMI is stratified into three categories according to World Health Organization (WHO) guidelines: normal weight (18.5–22.9 kg/m^2^), overweight (23.0–25.0 kg/m^2^), and obesity (greater than 25.0 kg/m^2^). Due to the absence of standardized thresholds for WHR, WHtR, and %BF, these metrics have been organized into quartiles for analysis.

Utilizing ANOVA, we observed that PreDM patients neared statistical significance across nearly all TyG and TyG-related indices in relation to BMI classifications—TyG (*p* = 0.05), TyG-BMI (*p* < 0.0001), and TyG-WHtR (*p* = 0.002). When PreDM individuals were categorized according to the WHR and WHtR quartiles, both TyG-WC and TyG-WHtR demonstrated statistically significant results (*p* < 0.0001).

Furthermore, %BF quartiles revealed significant distinctions for several indices, specifically TyG-BMI (*p* < 0.0001), TyG-WC (*p* = 0.04), and TyG-WHtR (*p* = 0.004).

In the T2DM cohort, ANOVA results indicated significant differences in the TyG and TyG-related indices across BMI categories—TyG (*p* = 0.02), TyG-BMI (*p* < 0.0001), TyG-WC (*p* < 0.0001), and TyG-WHtR (*p* < 0.0001). Notably, all quartiles of WHR, WHtR, and %BF displayed statistically significant findings for TyG-BMI, TyG-WC, and TyG-WHtR, each with *p*-values of <0.0001.

### 2.6. Connections of AIP, CMI, and CRR with BMI, WHR, WHtR, and %BF in the PreDM and T2DM Groups

Table 7 illustrates the interrelationships among the AIP, CMI, CRR, BMI, WHR, WHtR, and %BF in cohorts of individuals diagnosed with PreDM and T2DM.

BMI is classified into three categories based on WHO criteria: normal weight (18.5–22.9 kg/m^2^), overweight (23.0–25.0 kg/m^2^), and obesity (greater than 25.0 kg/m^2^). Due to the absence of established thresholds for the WHR, WHtR, and %BF, these metrics were delineated into quartiles for analysis.

In the PreDM cohort, no statistically significant differences were found across the WHtR and %BF quartiles. However, applying ANOVA for AIP and the Kruskal–Wallis test for the CMI and CRR revealed significant differences in AIP (*p* = 0.001), CMI (*p* < 0.0001), and CRR (*p* = 0.01) across BMI categories in the T2DM cohort. In the PreDM group, significance was only noted for the AIP (*p* = 0.01).

Both PreDM and T2DM groups demonstrated significant differences in AIP and CMI values when classified according to WHR quartiles, with the AIP (*p* = 0.04) and CMI (*p* = 0.02) showing significance in the PreDM cohort, while the T2DM cohort exhibited significance in the AIP (*p* = 0.001) and CMI (*p* = 0.03).

Moreover, the Kruskal–Wallis test indicated that the CMI achieved statistical significance when stratified by WHtR quartiles in patients with T2DM.

### 2.7. Correlations Between Metabolic Syndrome-Related Indices

#### 2.7.1. Correlations Between Metabolic Syndrome-Related Indices Concerning PreDM and T2DM Patients

Refer to Figure 2, where Pearson’s correlation analysis indicated that for the PreDM cohort, the values of TyG exhibited a significantly strong correlation with TG (*rho* = 0.900, *p*-value = 0.0001) and AIP (*rho* = 0.880, *p*-value = 0.0001); moreover, it showed a significantly moderate correlation with the CMI (*rho* = 0.760, *p*-value = 0.001) and TG/HDL-c (*rho* = 0.780, *p*-value = 0.001). TyG reached weak correlations with TyG-related indices, FPG, CRR, and %BF. For both cohorts, all of the TyG-related indices correlated with their composite parameters.

Regarding the T2DM group, as shown in Figure 3, TyG correlated strongly with AIP (r = 0.838, *p* < 0.0001). TyG showed moderate, statistically significant correlations with TG (*rho* = 0.742, *p*-value = 0.0001), CRR (*rho* = 0.575, *p*-value = 0.0003), CMI (*rho* = 0.676, *p*-value = 0.0001), and FPG (*rho* = 0.511, *p*-value = 0.01).

In the group with T2DM, our study identified a strong and statistically significant positive correlation between AIP and TG/HDL-c ratio (*rho* = 0.814, *p*-value < 0.0001). Moreover, moderate and statistically significant correlations of AIP were observed with the CMI (*rho* = 0.767, *p*-value = 0.002), CRR (*rho* = 0.732, *p*-value = 0.0001), and TG (*rho* = 0.718, *p*-value = 0.001). Furthermore, we identified a negative moderate correlation that was statistically significant between AIP and HDL-c (*rho* = −0.579, *p*-value = 0.001).

In the PreDM group, our study demonstrated a moderate and statistically significant positive correlation between AIP and CRR (*rho* = 0.501, *p*-value = 0.001). Furthermore, the AIP values exhibited strong positive correlations with the CMI (*rho* = 0.861, *p*-value = 0.02), TG/HDL-c ratio (*rho* = 0.869, *p*-value < 0.0001), and TG (*rho* = 0.862, *p*-value = 0.003). We identified a negative moderate correlation, which was statistically significant, between AIP and HDL-c (*rho* = −0.617, *p*-value < 0.0001).

Nonetheless, in the PreDM cohort, the CMI showed statistically significant, strong positive correlations between the CMI and TG (*rho* = 0.910, *p*-value = 0.002) and TG/HDL-c ratio (*rho* = 0.930, *p*-value = 0.0003), while with HDL-c it was a negative, weak one (*rho* = 0.240, *p*-value = 0.016).

In parallel, the T2DM cohort also presented a strong and statistically significant correlation between the CMI and TG/HDL-c ratio (*rho* = 0.979, *p*-value < 0.0001) and a weak negative one with HDL-c (*rho* = 0.334, *p*-value = 0.005).

In the PreDM cohort, %BF presented moderate, statistically significant correlations with weight (*rho* = 0.746, *p*-value = 0.05) and BMI (*rho* = 0.783, *p*-value = 0.009); while the T2DM group showed BMI (*rho* = 0.747, *p*-value = 0.0001) and TyG-BMI (*rho* = 0.711, *p*-value < 0.0001). Additionally, it correlated weakly with TyG-WC (*rho* = 0.327, *p*-value = 0.0002), TyG-WHtR (*rho* = 0.491, *p*-value = 0.0001), WHtR (*rho* = 0.483, *p*-value < 0.0001), and weight (*rho* = 0.397, *p*-value < 0.0001). The %BF for T2DM patients also correlated negatively, but moderately with height (*rho* = −0.490, *p*-value < 0.0001).

#### 2.7.2. Correlations Between Metabolic Syndrome-Related Indices Concerning PreDM Patients After the MUHNW and MUHO Phenotypes

The majority of the identified correlations were found to remain consistent when we investigated the parameters after the MUHNW and MUHO phenotypes. See Figure 4 and Figure 5.

The most notable correlations were the following:-PreDM MUHNW:
CMI correlated positively, strongly with AIP (*rho* = 0.856, *p*-value < 0.0001);TG/HDL-c correlated positively and strongly with the CMI (*rho* = 0.992, *p*-value = 0.0001);TyG-WHtR correlated weakly with the CMI (*rho* = 0.314, *p*-value of 0.007).
-PreDM MUHO:
AIP presented statistically significant correlations with the CMI (*rho* = 0.870, *p*-value = 0.0001), and TG/HDL-c ratio (*rho* = 0.874, *p*-value < 0.0001)—strong and positive;TG/HDL-c correlated positively and strongly with the CMI (*rho* = 0.991, *p*-value = 0.0001);CRR correlated moderately with the CMI (*rho* = 0.638, *p*-value < 0.0001).


#### 2.7.3. Correlations Between Metabolic Syndrome-Related Indices Concerning PreDM and T2DM Patients for the MUHNW and MUHO Phenotypes

The majority of the identified correlations were found to remain consistent, with even stronger ones when we investigated the parameters after the MUHNW and MUHO phenotypes. See Figure 6 and Figure 7.

The most notable correlations were:-T2DM MUHNW:
AIP presented statistically significant correlations with the CMI (*rho* = 0. 904, *p*-value = 0.0001)—strong and positive;The CMI correlated positively and strongly with TG/HDL-c (*rho* = 0.964, *p*-value = 0.0001);The CRR correlated strongly with AIP (*rho* = 0.802, *p*-value < 0.0001).
-T2DM MUHO:
AIP presented statistically significant correlations with the CRR (*rho* = 0.716, *p*-value = 0.0001)—moderate and positive;The CMI correlated positively and strongly with TG/HDL-c (*rho* = 0.983, *p*-value = 0.0001);The CRR correlated positively and moderately with TG/HDL-c (*rho* = 0.646, *p*-value = 0.0001).


### 2.8. Comparative Analysis of Metabolic Syndrome-Related Indices Among Subgroups of Females and Males in PreDM and T2DM Groups

In our analysis, we stratified the PreDM and T2DM cohorts by gender, focusing on the male and female groups separately.

Among the male participants, we found significant differences when comparing the PreDM group to the T2DM cohort in various metabolic parameters. Specifically, metrics such as the TyG showed *p* < 0.0001, TyG-WC reached *p* = 0.05, the TG/HDL-c had *p* = 0.0001, AIP was noted at *p* = 0.004, and the CMI presented a *p*-value of 0.009.

In the female population, the comparison between the PreDM and T2DM groups highlighted significant associations as well.

Key findings included a *p*-value of less than 0.0001 for TyG, *p* = 0.0007 for TyG-BMI, *p* = 0.01 for the CRR, and *p* = 0.0003 for body fat percentage (%BF).

These results, shown in Figure 8 and Table 8, underscore the differences in metabolic dysregulation between the PreDM and T2DM states across genders, revealing critical insights into the underlying pathophysiological processes.

### 2.9. Diagnostic Accuracy of Different Indices and Biomarkers

Our study focused on assessing the ability of specific parameters (AIP, TG/HDLc, CMI, WHtR, CRR, %BF, and TyG-WHtR) to track the progression of MetS in patients with PreDM and T2DM through the receiver operating characteristic (ROC) curve analysis. Additionally, we evaluated the ROC curves corresponding to various phenotypes of cardiometabolic syndrome. For each parameter, we identified the optimal cut-off value by maximizing the combined sensitivity and specificity. The results are summarized in Table 9 and illustrated in Figure 9A–U, which presents the ROC curves for the parameters under investigation.

When we compared between the PreDM and T2DM cohorts by analyzing the area under the ROC curve (AUC) obtained for the evaluated parameters, we noticed that the indices related to the MetS that we investigated showed a good performance, especially AIP (AUC of 0.623, sensitivity of 54%, specificity of 67%, cut-off of 0.49), followed by the CMI (AUC of 0.592, sensitivity of 54%, specificity of 59%, cut-off of 1.71), and lastly the CRR (AUC of 0.577, sensitivity of 53%, specificity of 59%, cut-off of 3.79). It is of interest that the %BF (AUC of 0.527, sensitivity of 53.50%, specificity of 54%, cut-off of 40.05) and TyG-WHtR (AUC of 0.517, sensitivity of 50%, specificity of 64%, cut-off of 5.56) did not show better values than the ones mentioned.

In contrast, for the MUHNW phenotype, we obtained one of the best values for TyG-WHtR (AUC of 0.765, sensitivity of 73.08%, specificity of 69.70%, cut-off of 4.78). The unexpected result was also maintained for the CRR (AUC of 0.549, sensitivity of 42.31%, specificity of 63.64%, cut-off of 3.46), CMI (AUC of 0.547, sensitivity of 57.69%, specificity of 51.52%, cut-off of 1.39), and AIP (AUC of 0.533, sensitivity of 55.77%, specificity of 51.52%, cut-off of 0.37). The worst values were obtained for %BF (AUC of 0.506, sensitivity of 57.69%, specificity of 48.48%, cut-off of 29.14). For the MUHO phenotype, AIP reached an AUC of 0.659, a sensitivity of 67.57%, and a specificity of 64.18%, with a cut-off of 0.65; the CMI (AUC of 0.641, sensitivity of 65.54%, specificity of 61.19%, cut-off of 1.71) and CRR (AUC of 0.591, sensitivity of 64.19%%, specificity of 58.21%, cut-off of 3.87) showed lower values. It is mentioned that %BF attained an AUC of 0.607, a sensitivity of 51.53%, a specificity of 50.75%, and a cut-off value of 42.55.

## 3. Discussion

This study investigated the metabolic and cardiometabolic profiles of individuals with PreDM and T2DM, with an emphasis on the utility of metabolic syndrome-related indices—TyG, TyG-composed indices, AIP, CMI, and CRR—in assessing cardiovascular risk and phenotype distinctions. Our data demonstrate that patients with T2DM exhibit significantly worsened metabolic, inflammatory, and anthropometric profiles compared to PreDM patients, reflecting the progressive nature of metabolic dysregulation. We acknowledge the inherent overlap in defining phenotypes and risk indices; however, the focus was on identifying phenotype-specific trends in validated surrogate risk markers to inform early preventive strategies.

The TyG index has demonstrated notable cost-effectiveness and reproducibility, positioning it as a promising biomarker for forecasting cardiovascular events in individuals diagnosed with T2DM and MetS [32,33]. The findings in our study were in accordance with the ones stated above.

A recent study demonstrated that both the TyG index and the AIP serve as independent prognostic factors for major adverse cardiovascular and cerebrovascular events (MACCEs) in patients with acute coronary syndrome (ACS) undergoing percutaneous coronary intervention (PCI). While the predictive capabilities of the TyG index and AIP for MACCEs were comparable, their combined predictive model did not yield significant enhancements in risk stratification. The AUC for the AIP was 0.626. The optimal threshold value identified was 0.30 for the AIP [34,35]. Our study found, for the MUHO phenotype, an AUC of 0.757, a cut-off of 9.11, and the AIP reached an AUC of 0.659, with a cut-off of 0.65.

Individuals with T2DM exhibited increased prevalence of comorbid conditions, including hypertension, dyslipidemia, and hepatosteatosis. These findings align with the existing literature highlighting the multisystem consequences of insulin resistance and chronic hyperglycemia [36,37]. The elevated prevalence of alcohol consumption and tobacco use within this demographic corresponds with lifestyle behaviors that are recognized to intensify oxidative stress and the inflammatory processes implicated in the advancement of MetS [38]. Our study showed consistent findings, with many patients having these types of behaviors.

Two studies emphasized that the TyG index, a surrogate marker of insulin resistance, was significantly higher in the T2DM group compared to the PreDM group, corroborating findings from recent studies emphasizing its superior diagnostic value over the homeostatic model assessment for insulin resistance (HOMA-IR) in metabolic disorders [39,40]. The TyG index and its adjusted indices—TyG-BMI, TyG-WC, and TyG-WHtR—showed statistically significant differences, particularly in T2DM individuals. These associations reflect the compounded metabolic burden in patients exhibiting obesity with or without overt IR.

Our results align with a growing body of evidence suggesting that TyG and its related indices are powerful indicators of both glucose metabolism disorders and atherosclerotic risk [41,42]. Moreover, their association with obesity-related indices such as BMI, WHR, and %BF further reinforces their utility in stratifying metabolic risk, especially among MUHO individuals. A meta-analysis by Wang et al. concluded that TyG-related indices could independently predict MetS and its components across different BMI categories, supporting their use in primary care for early intervention [43].

AIP, defined as the log-transformed ratio of TG to HDL-c, was significantly higher in the T2DM cohort, particularly among the MUHO phenotype. This finding supports the hypothesis that AIP reflects atherogenic lipoprotein particle size and serves as a surrogate for small dense LDL particles [44]. In our study, AIP was significantly associated with WHR and BMI in both PreDM and T2DM cohorts, confirming previous studies that highlight its role in predicting cardiovascular events and subclinical atherosclerosis [45].

Additionally, we found significant correlations between AIP and obesity indices in the T2DM group, but less so in the PreDM group, indicating that AIP may have a threshold effect, becoming more predictive once overt diabetes and related dyslipidemia manifest. These results support recommendations for the use of AIP in diabetes risk stratification, as discussed in recent clinical evaluations [46].

The CMI, which incorporates the TG/HDL-c ratio and WHtR, demonstrated significant differences between MUHNW and MUHO phenotypes in the T2DM cohort, and was positively correlated with BMI and WHR quartiles. This confirms the utility of the CMI as a dual marker of visceral adiposity and lipid imbalance, which was also reported in a 2021 study by Liu et al. [46]; the CMI is particularly valuable in populations where traditional markers such as BMI may underestimate metabolic risk, especially in normal-weight but metabolically unhealthy individuals.

In our cohort, the CMI correlated significantly with other lipid indices and anthropometric measures, consistent with data from the literature showing that the CMI strongly predicts MetS and T2DM incidence independent of age and sex [47,48].

The CRR (TC/HDL-c) values were significantly elevated in the T2DM group, particularly within the MUHO phenotype, echoing past findings that elevated CRR is associated with increased cardiovascular morbidity [49]. The significant correlations with obesity indices such as the WHR and WHtR reinforce the interplay between abdominal obesity and dyslipidemia in shaping cardiovascular risk profiles [50].

When dissecting the MUHNW and MUHO phenotypes, significant variations were noted in TyG, AIP, CMI, and CRR across both the PreDM and T2DM cohorts. Notably, T2DM-MUHO individuals exhibited the most pronounced elevations in all indices, suggesting a clustering of metabolic and inflammatory risk factors. These results are supported by studies showing that metabolically healthy obese individuals have a disproportionately lower risk of cardiovascular events and insulin resistance progression, unlike the results of our study, where the MUHO phenotype cohort presented elevated cardiovascular-, obesity-, and MetS-related indices [51].

Furthermore, gender-based analysis revealed higher cardiometabolic risk indices in males, particularly in the T2DM group—a pattern consistent with findings that men tend to accumulate more visceral fat, a key driver of insulin resistance and metabolic dysfunction [52].

Our results of the different MetS-related indices may also be tied to nutritional status, which might be of importance in the cardiometabolic phenotypes. A study showed that low vitamin C levels observed in diabetic patients with periodontal disease suggest a shared inflammatory and oxidative stress pathway. This might support the link between micronutrient deficiency and the metabolic disturbances seen in the cardiometabolic phenotypes of diabetes [53].

While this study provides meaningful contributions to the understanding of metabolic indices and cardiovascular risk, several limitations must be taken into account when interpreting the results. This cross-sectional and retrospective design inherently restricts the ability to conclude temporal sequences or causality between the observed variables. Without longitudinal data, it remains unclear whether elevated metabolic indices precede cardiovascular risk or simply coexist with it. Furthermore, the study was carried out at a single clinical center and involved participants from a relatively uniform Romanian population, which may affect the external validity of the findings. Populations with different genetic backgrounds, lifestyle patterns, or healthcare access may exhibit distinct metabolic and cardiovascular profiles, thus limiting the applicability of our results across broader demographic or ethnic groups.

Another limitation lies in the use of surrogate biomarkers—such as the TyG, AIP, CMI, and CRR—to estimate IR and cardiovascular risk. Although these markers have been validated in prior research and are useful for large-scale epidemiological studies, they cannot fully substitute direct clinical measures or hard outcomes, such as glucose clamp tests or incident cardiovascular events. Additionally, potential confounding factors such as medication usage, dietary intake, physical activity, psychosocial stress, environmental exposures, and detailed pharmacological profiles were not comprehensively accounted for due to constraints in the dataset. These unmeasured variables may influence both metabolic parameters and risk scores, potentially biasing the associations reported.

While we considered conducting a sensitivity analysis to exclude participants who experienced cardiovascular events in the near term, the necessary data were not available, preventing such an approach. This further emphasizes the need for studies with more detailed clinical follow-up. Moreover, the observational nature of the study precludes the establishment of any directional or mechanistic insight between the metabolic phenotype and cardiometabolic outcomes.

Finally, given the single-center design and relatively narrow participant base, caution is warranted when attempting to generalize these results to other settings or populations. Replication in multi-center, multi-ethnic, or nationally representative cohorts is essential to validate these findings and enhance their translational relevance. Future studies should adopt prospective designs with comprehensive data collection to allow for more robust adjustment for confounding variables, direct measurement of insulin resistance and cardiovascular endpoints, and ultimately, a clearer understanding of causal pathways.

## 4. Materials and Methods

Interventional research involving both human and animal subjects, as well as any study necessitating ethical approval, must disclose the approving authority along with the relevant ethical approval code. The study was conducted following the principles outlined in the Declaration of Helsinki and received ethical clearance from the Ethics Committee of the Filantropia Municipal Clinical Hospital (approval number 887/15, January 2024), situated in Dolj, Romania. The primary aim was to investigate the relationship between defined cardiometabolic phenotypes and a range of validated cardiovascular risk indices—TyG, AIP, CMI, and CRR—in individuals with prediabetes and T2DM. This included examining how lifestyle, biochemical, and anthropometric factors covaried within these phenotypes.

### 4.1. Patient Selection, Review of Clinical History, Evaluation of Biometric Metrics, and Compilation of Demographic Information

The study’s inclusion criteria required participants to be 18 years or older with a diagnosis of T2DM or PreDM, consistent with the MetS diagnostic criteria. Participants were recruited from the Outpatient Departments specializing in Diabetes, Nutrition, and Metabolic Diseases at Filantropia Municipal Clinical Hospital in Craiova. Each individual provided informed consent before their voluntary involvement in the research.

We excluded individuals diagnosed with diabetes mellitus other than T2DM, those presenting with autoimmune diseases as comorbidities, and patients with missing critical data, specifically waist circumference and body mass index, as these parameters were considered vital for the integrity of our study.

Subjects with chronic microvascular complications associated with T2DM, including diabetic peripheral polyneuropathy, diabetic nephropathy, and diabetic retinopathy, were excluded from the study cohort. Diabetic retinopathy (DR) was diagnosed utilizing a comprehensive dilated fundus examination [54,55]. Following ADA recommendations, the evaluation of diabetic peripheral neuropathy encompasses the assessment of both temperature sensation—indicative of small fiber pathology—and vibration sensation—reflective of large fiber integrity. Additionally, this assessment includes the identification of characteristic clinical manifestations such as pain, dysesthesia, and sensory numbness [54,56]. We assessed chronic kidney disease (CKD) following the Kidney Disease Improving Global Outcomes (KDIGO) criteria: CKD was defined by a urinary albumin-to-creatinine ratio (UACR) greater than 30 mg/g and/or an eGFR below 60 mL/min/1.73 m^2^, as stipulated in the KDIGO 2021 Guidelines. Diabetic kidney disease (DKD), also known as diabetic nephropathy (DN), is identified by the coexistence of CKD and diabetes [57,58].

In the prediabetic cohort, we excluded individuals under the age of 18, pregnant women, those who had experienced an acute infection or inflammatory condition within the preceding month, individuals with chronic infections or inflammatory disorders, and patients diagnosed with cancer.

We executed a non-interventional, cross-sectional epidemiological study over a six-month period, enrolling 324 consecutive patients newly diagnosed with T2DM. A control group comprised 144 PreDM patients, matched for age, gender ratio, and urban versus rural demographic status.

The primary objective of the study was to amass exhaustive data concerning various health and lifestyle parameters. This encompassed anthropometric measurements, clinical variables, and laboratory test outcomes, along with demographic and lifestyle information, all acquired via a structured interview questionnaire. This cross-sectional analysis evaluated surrogate indices rather than direct cardiovascular outcomes. No incident CVD endpoints or follow-up data were included, and therefore, this study should not be interpreted as predictive of clinical events. The study was designed as an exploratory association analysis and does not imply prognostic prediction of cardiovascular events. Future studies with clinical endpoints and follow-up are required to validate these associations.

Although we focused on surrogate indices rather than clinical diagnoses, the term ‘cardiovascular disease risk’ in this study refers broadly to conditions commonly associated with MetS, including hypertension, atherosclerotic cardiovascular disease, coronary artery disease, and cerebrovascular events, as implied by the indices studied.

Demographic variables examined included age, sex, monthly household income, and educational attainment. Health- and lifestyle-related factors assessed included a history of tobacco use or alcohol consumption, familial predisposition to hypertension, diabetes mellitus, and cardiovascular diseases, as well as the weekly duration of deliberately undertaken moderate physical activity. We recognize the absence of dietary intake, psychosocial stress measures, and genetic predisposition markers in our dataset, which are known modifiers of cardiometabolic risk. Furthermore, the lack of detailed data on medication type, dosage, and adherence limits our ability to fully adjust for pharmacologic effects on metabolic indices.

### 4.2. Assessment of T2DM and PreDM Patients

Prediabetes is characterized by one or more of the following parameters: (1) a clinical diagnosis made by a qualified healthcare provider; (2) an HbA1c concentration greater than 5.7% but less than 6.5%; (3) an FPG level between 5.6 mmol/L and 7.0 mmol/L; or (4) a 2 h plasma glucose measurement from an oral glucose tolerance test (OGTT) that lies in the range of 7.8 mmol/L to 11.0 mmol/L [59]. Individuals exhibiting obesity, especially in the abdominal or visceral regions, alongside dyslipidemia marked by increased triglyceride levels and/or decreased HDL-c, in conjunction with hypertension, were included in the prediabetes cohort group, provided that they met the specified inclusion criteria.

A diagnosis of diabetes is established when one or more of the following criteria are fulfilled: confirmation by a healthcare professional; an HbA1c level of 6.5% or higher; a FPG measurement at or above 7.0 mmol/L; a random blood glucose level of 11.1 mmol/L or more; a OGTT blood glucose level exceeding 11.1 mmol/L; or the presence of hyperglycemic symptoms—such as polyuria, polydipsia, or significant weight loss—accompanied by a random glucose measurement indicative of a hyperglycemic crisis [59].

### 4.3. MetS Definition and Cardiometabolic Phenotypes

Cardiometabolic phenotypes were categorized based on predefined combinations of BMI and metabolic health markers, prior to any calculation of risk indices. Groups were defined independently of the indices.

MetS was classified based on the harmonized definition established in 2009, requiring the presence of at least three out of five specified diagnostic criteria for diagnosis (criteria mentioned in Table 10) [60].

We computed the BMI of the participants, utilizing their height and weight measurements. The calculation adheres to the formula BMI = weight (kg)/height^2^ (m^2^) [61]. Patients diagnosed with stable coronary heart disease exhibited a progressive elevation in cardiometabolic and inflammatory risk markers as their BMI surpassed 25 kg/m^2^ [62].

We divided the patients into 4 cardiometabolic phenotypes in Table 11, adapted based on Soheilifard et al. [28].

Participants were stratified into four cardiometabolic phenotypes based on BMI (threshold: 25 kg/m^2^) and the presence of ≥3 MetS criteria. This classification served solely as a grouping framework for comparative analysis. A flowchart illustrating this stratification process and the subsequent evaluation of independent cardiometabolic risk indices (TyG, AIP, CMI, and CRR) is presented in Figure 10. After applying the criteria, we obtained the following cohorts after the classification of cardiometabolic phenotypes. Given the insufficient sample size for the MHO PreDM and MHNW T2DM cohorts, our comparative analysis was limited to the MUHNW and MUHO phenotypes. Therefore, we excluded 33 patients from the PreDM cohort and 124 patients from the T2DM cohort. Consequently, a total of 200 out of 324 patients diagnosed with T2DM and 100 out of 144 patients with PreDM successfully completed the study and were incorporated into the final analysis.

### 4.4. Evaluation of Various MetS-Related Indices (TyG, TyG-Related Indices, AIP, CMI, and CRR)

Circumferential measurements were obtained at the femoral trochanters for HC and at the midpoint between the upper iliac crest and the lower rib cage for WC. The WHR was computed using the formula WC (cm)/HC (cm), serving as an indicator of abdominal obesity. Furthermore, visceral adiposity was evaluated using the WHtR, derived from the formula WC (cm)/height (cm).

We partitioned WHR, WHtR, BAI, and VAI into four quarters due to the absence of standard categories.

AIP, CMI, and CRR were determined after the following formulas [11,63,64,65]:




$AIP=log10[\frac{TG}{HDL-C}]$



$CMI=\left[\frac{TG}{HDL-C}\right]\times WHtR$



$CRR=[\frac{TC}{HDL-C}]$




TyG and TyG-derived indices were calculated with the following formulas [66,67,68]:



$TyG=ln[\frac{TG\times FPG}{2}]$

TyG-BMI = TyG × BMITyG-WHtR = TyG × WHtRTyG-WC = TyG × WC

### 4.5. Laboratory Investigations

After obtaining the anthropometric measurements, we proceeded to conduct more in-depth assessments with the subjects in the laboratory setting.

We utilized a clinical chemistry analyzer (ARCHITECT C4000, Abbott, Abbott Park, IL, USA) to evaluate the laboratory data.

Utilizing flow cytometry under Coulter’s principle, we successfully established an extended leukocyte differential by analyzing five distinct parameters using the MINDRAY BC-6800 (Mindray, Shenzhen, China). This approach enabled us to identify and characterize a range of hemoleucogram markers effectively.

Serum creatinine concentrations were assessed, and the estimated eGFR was determined utilizing the Modification of Diet in Renal Disease (MDRD) formula [69].

### 4.6. Statistical Analysis

We utilized Microsoft Excel to process and manage patient data extracted from medical records. For subsequent data analysis, we employed GraphPad Prism version 10.3.1 (LLC, San Diego, CA, USA). The normality of the data was evaluated using both the Kolmogorov–Smirnov test and the Shapiro–Wilk test to determine adherence to a normal distribution.

The means and SD for the variables, including age, HbA1c, FPG, 2h-PG, eGFR, WBC, HBG, PLT, SBP, DBP, height, WHR, %BF, TC, LDL-c, HDL-c, TyG, TyG-BMI, TyG-WC, TyG-WHtR, and AIP, were found to follow normal distributions. Conversely, the variables such as ESR, CRP, AST, ALT, creatinine, weight, WC, HC, WHtR, BMI, TG, TG/HDL-c, CMI, and CRR exhibited non-normal distributions; hence, these results are reported as medians with interquartile ranges. Categorical variables are presented as percentages.

We assessed continuous variables using one-way ANOVA or the Kruskal–Wallis test, appropriate for non-Gaussian distributions, to detect inter-group differences. For categorical variables, we utilized the χ^2^ test for analysis.

Pearson’s correlation coefficients (−1 < ρ < 1) were utilized to evaluate significant correlations between the measured levels of height, weight, BMI, WC, HC, WHR, WHtR, HbA1c, FPG, 2h-PG, TC, LDL-c, HDL-c, TG, SBP, DBP, TyG, TyG-BMI, TyG-WC, TyG-WHtR, TG/HDL-c, AIP, CMI, %BF, and CRR.

Receiver operating characteristic (ROC) curves were utilized to evaluate the sensitivity and specificity of a range of parameters, including AIP, TG/HDLc, CMI, WHtR, CRR, %BF, and TyG-WHtR.

## 5. Conclusions

This study highlights the intricate interplay between cardiometabolic phenotypes and cardiovascular risk indices in individuals with prediabetes and T2DM. Our findings underscore the significant distinctions in metabolic and cardiovascular profiles among the MUHNW and MUHO phenotypes, providing compelling evidence that cardiometabolic risk is not solely dependent on BMI, but also on metabolic health status. These phenotypes are strongly associated with higher cardiometabolic risk indices, suggesting a potential role in early risk stratification, pending validation in prospective studies.

Key indicators such as the TyG index and its derivatives, along with AIP, CMI, and CRR, showed strong associations with both traditional anthropometric markers (BMI, WHR, WHtR, and %BF) and metabolic derangements. These indices were significantly elevated in T2DM patients, particularly those classified within the MUHO phenotype, suggesting a higher cardiovascular risk burden.

Importantly, we demonstrated that individuals with T2DM exhibit more pronounced metabolic disturbances than those with prediabetes, regardless of phenotype. This supports the value of early identification and targeted management strategies tailored not just based on weight status, but based on a comprehensive metabolic assessment.

Our results advocate for the integration of advanced indices, such as AIP, CMI, CRR, and %BF, into routine clinical evaluations to enhance cardiovascular risk stratification. This is especially pertinent for identifying high-risk individuals within the ostensibly “normal weight” population who may be metabolically unhealthy.

Future research should aim to validate these findings in longitudinal cohorts and diverse populations and assess the utility of these indices in predicting clinical outcomes, thereby refining risk-based prevention and treatment paradigms in metabolic and cardiovascular health. Also, future research should employ prospective, multicentric designs with robust data on lifestyle, medications, psychosocial exposures, and follow-up for cardiovascular outcomes. We considered performing a sensitivity analysis excluding participants likely to develop cardiovascular events within the near term; however, due to the retrospective nature and lack of follow-up data, this was not feasible. Future prospective cohorts should incorporate such analyses.

## Figures and Tables

**Figure 1 ijms-26-06227-f001:**
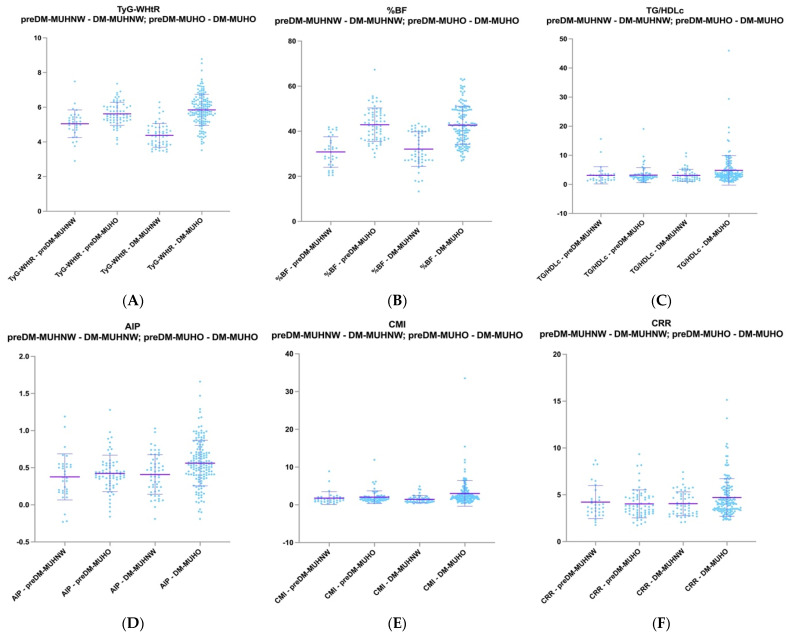
Representation of the distribution of different MetS-related indices for MUHNW and MUHO phenotypes for PreDM and T2DM patients for the following: TyG-WHtR (**A**), %BF (**B**), TG/HDL-c (**C**), AIP (**D**), CMI (**E**), and CRR (**F**).

**Figure 2 ijms-26-06227-f002:**
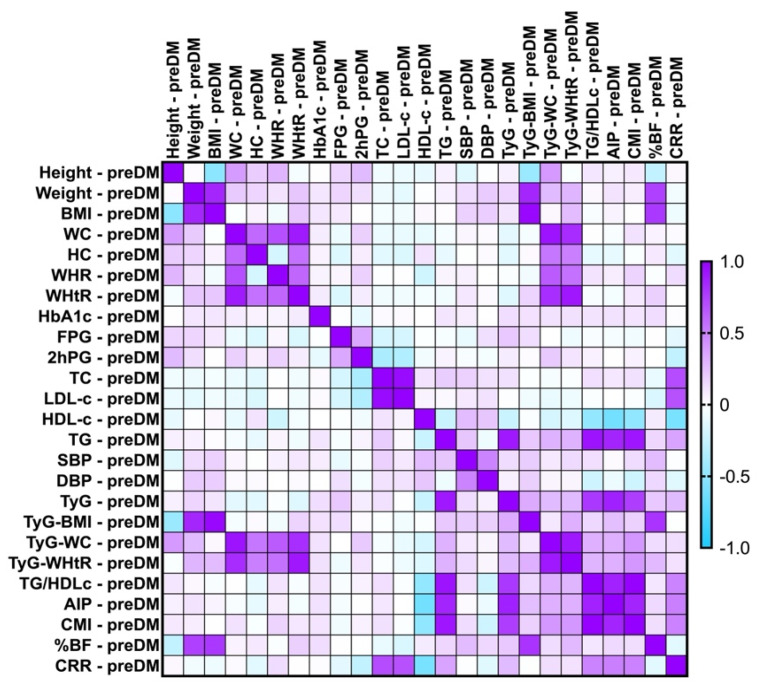
Correlation matrix between glycemic spectrum, lipid spectrum, anthropometric indices, and metabolic syndrome-related indices in the PreDM cohort. The correlation heatmap illustrates the relationships between the measured indicators. Strong positive correlations are represented by intense purple, while strong negative correlations are depicted in bright blue.

**Figure 3 ijms-26-06227-f003:**
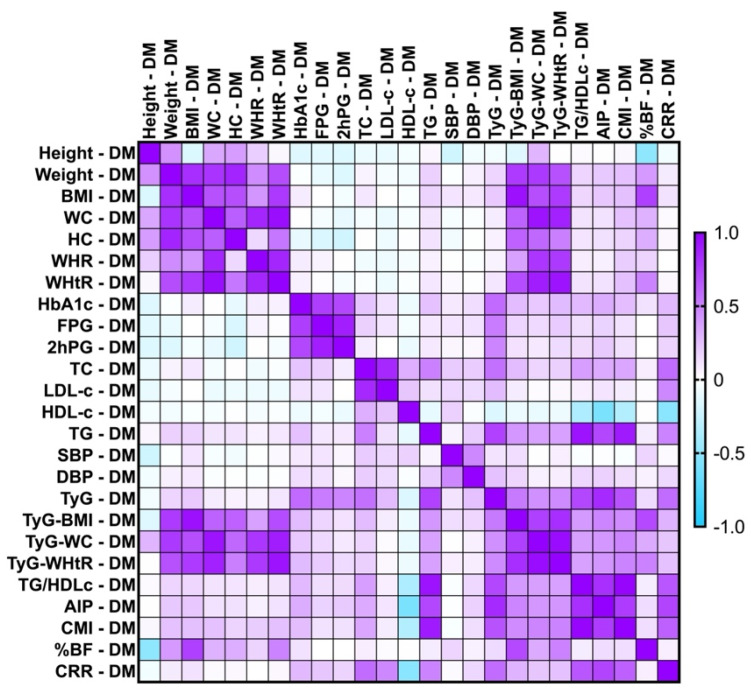
Correlation matrix between glycemic spectrum, lipid spectrum, anthropometric indices, and metabolic syndrome-related indices in the T2DM cohort. The correlation heatmap illustrates the relationships between the measured indicators. Strong positive correlations are represented by intense purple, while strong negative correlations are depicted in bright blue.

**Figure 4 ijms-26-06227-f004:**
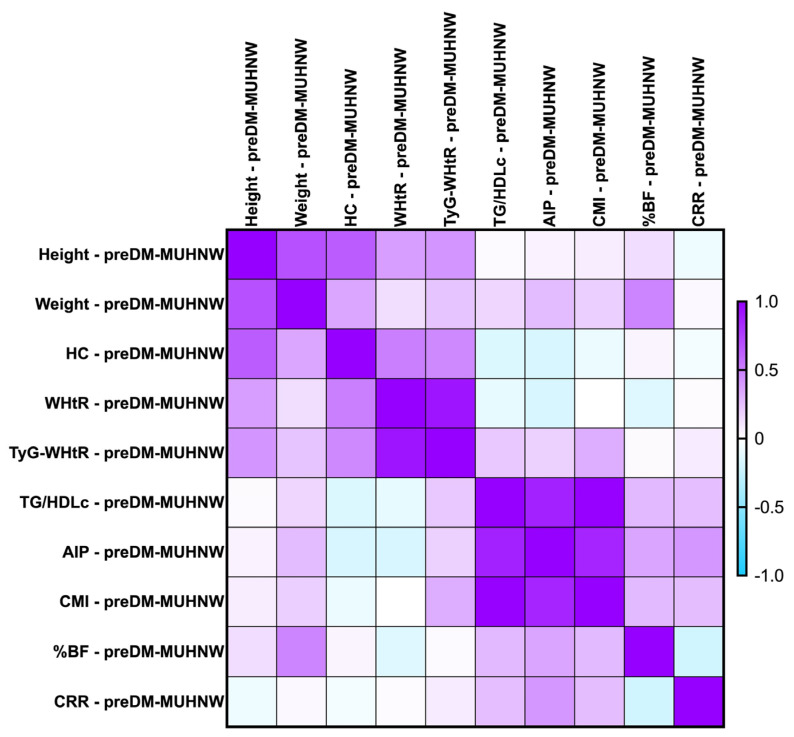
Correlation matrix between anthropometric indices and metabolic syndrome-related indices in the PreDM cohort, classified in the MUHNW phenotype. The correlation heatmap illustrates the relationships between the measured indicators. Strong positive correlations are represented by intense purple, while strong negative correlations are depicted in bright blue.

**Figure 5 ijms-26-06227-f005:**
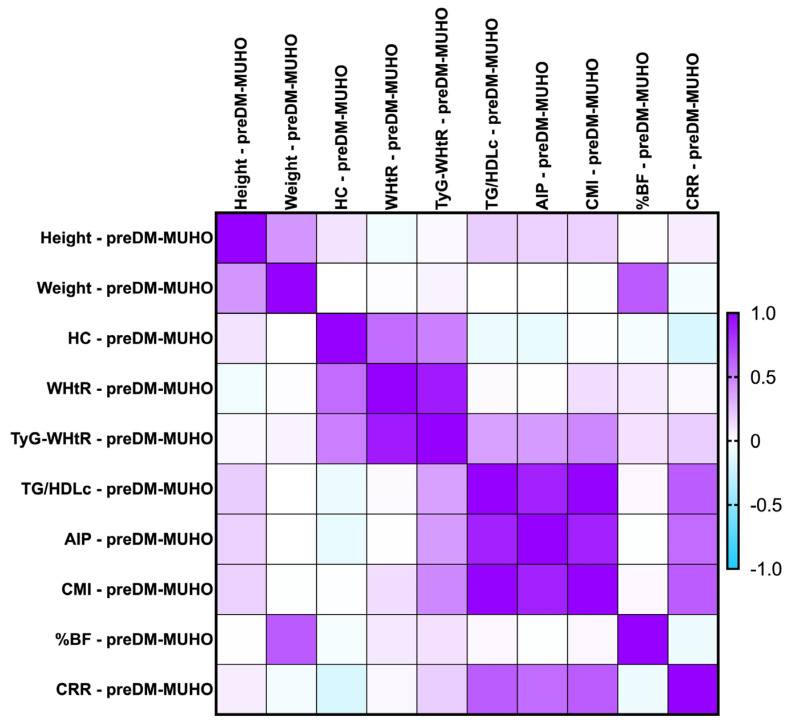
Correlation matrix between anthropometric indices and metabolic syndrome-related indices in the PreDM cohort, classified in the MUHO phenotype. The correlation heatmap illustrates the relationships between the measured indicators. Strong positive correlations are represented by intense purple, while strong negative correlations are depicted in bright blue.

**Figure 6 ijms-26-06227-f006:**
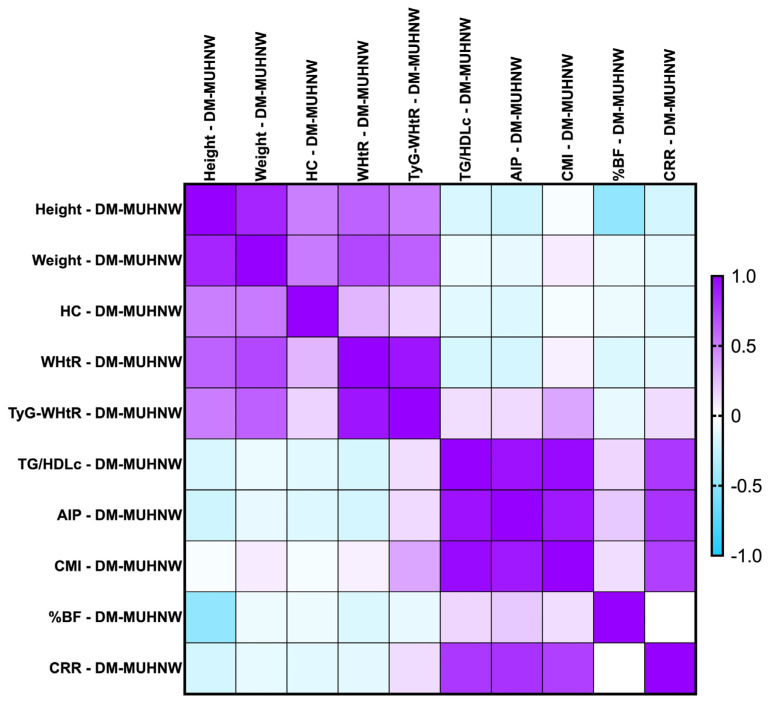
Correlation matrix between anthropometric indices and metabolic syndrome-related indices in the T2DM cohort, classified in the MUHNW phenotype. The correlation heatmap illustrates the relationships between the measured indicators. Strong positive correlations are represented by intense purple, while strong negative correlations are depicted in bright blue.

**Figure 7 ijms-26-06227-f007:**
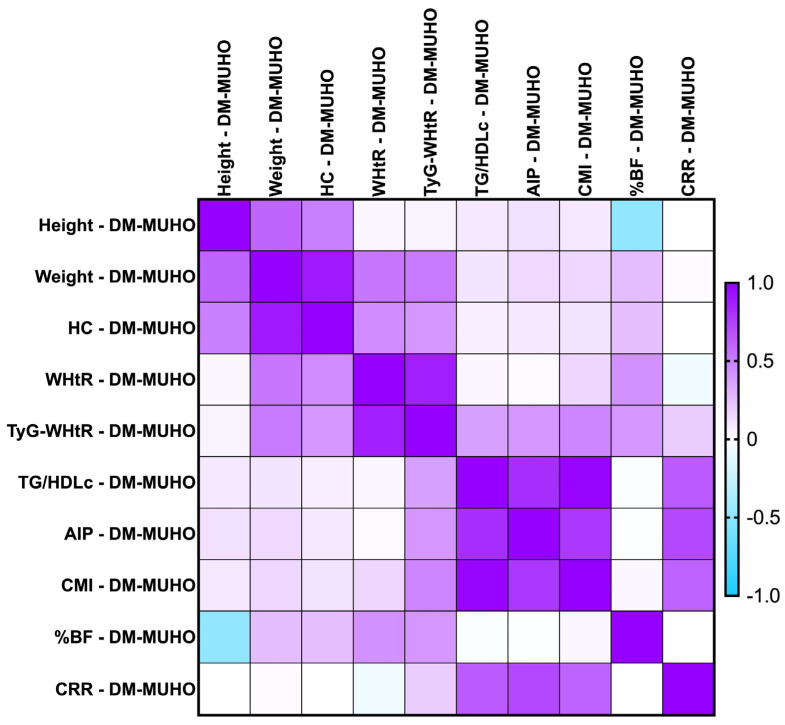
Correlation matrix between anthropometric indices and metabolic syndrome-related indices in the T2DM cohort, classified in the MUHO phenotype. The correlation heatmap illustrates the relationships between the measured indicators. Strong positive correlations are represented by intense purple, while strong negative correlations are depicted in bright blue.

**Figure 8 ijms-26-06227-f008:**
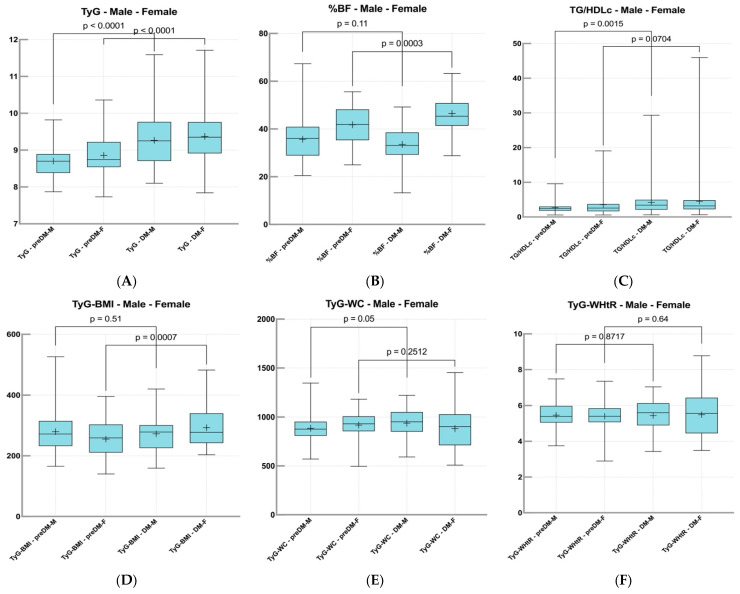
Representation of the comparison of MetS-related indices between male and female cohorts: TyG values for both PreDM and T2DM cohorts separated for male and female (**A**), %BF values for both PreDM and T2DM cohorts separated for male and female (**B**), TG/HDL-c ratio values for both PreDM and T2DM cohorts separated for male and female (**C**), TyG-BMI values for both PreDM and T2DM cohorts separated for male and female (**D**), TyG-WC values for both PreDM and T2DM cohorts separated for male and female (**E**), TyG-WHtR values for both PreDM and T2DM cohorts separated for male and female (**F**), AIP values for both PreDM and T2DM cohorts separated for male and female (**G**), CMI values for both PreDM and T2DM cohorts separated for male and female (**H**), and CRR values for both PreDM and T2DM cohorts separated for male and female (**I**).

**Figure 9 ijms-26-06227-f009:**
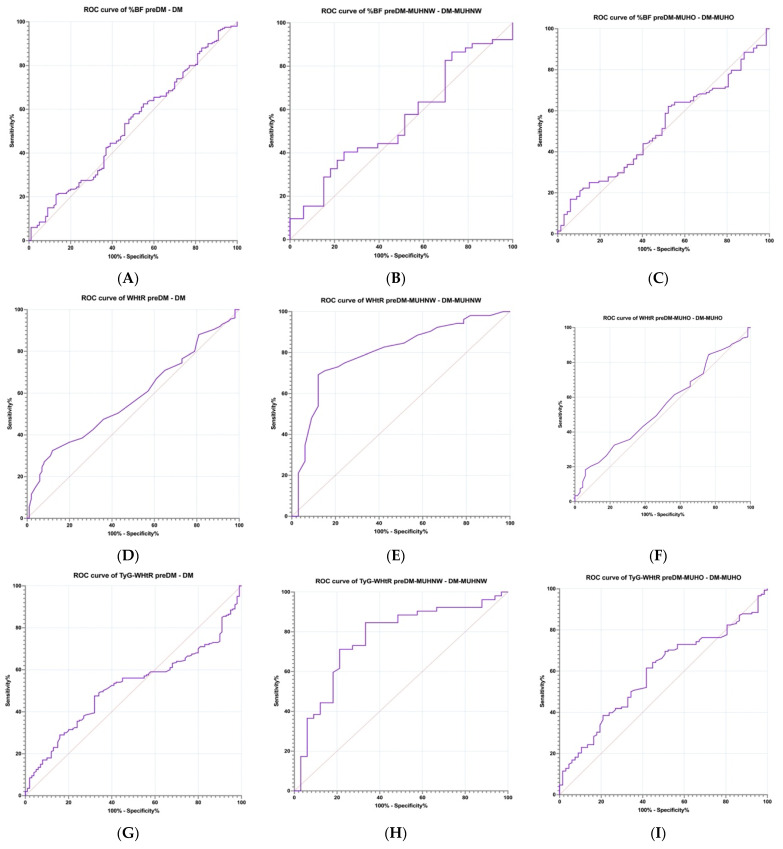
Receiver operating characteristic (ROC) curve for %BF PreDM—DM (**A**), %BF PreDM-MUHNW—DM-MUHNW (**B**), %BF PreDM-MUHO—DM-MUHO (**C**), WHtR PreDM—DM (**D**), WHtR PreDM-MUHNW—DM-MUHNW (**E**), WHtR PreDM-MUHO—DM-MUHO (**F**), TyG-WHtR PreDM—DM (**G**), TyG-WHtR PreDM-MUHNW—DM-MUHNW (**H**), TyG-WHtR PreDM-MUHO—DM-MUHO (**I**), TG/HDL-c PreDM—DM (**J**), TG/HDL-c PreDM-MUHNW—DM-MUHNW (**K**), TG/HDL-c PreDM-MUHO—DM-MUHO (**L**), AIP PreDM—DM (**M**), AIP PreDM-MUHNW—DM-MUHNW (**N**), AIP PreDM-MUHO—DM-MUHO (**O**), CMI PreDM—DM (**P**), CMI PreDM-MUHNW—DM-MUHNW (**Q**), CMI PreDM-MUHO—DM-MUHO (**R**), CRR PreDM—DM (**S**), CRR PreDM-MUHNW—DM-MUHNW (**T**), and CRR PreDM-MUHO—DM-MUHO (**U**).

**Figure 10 ijms-26-06227-f010:**
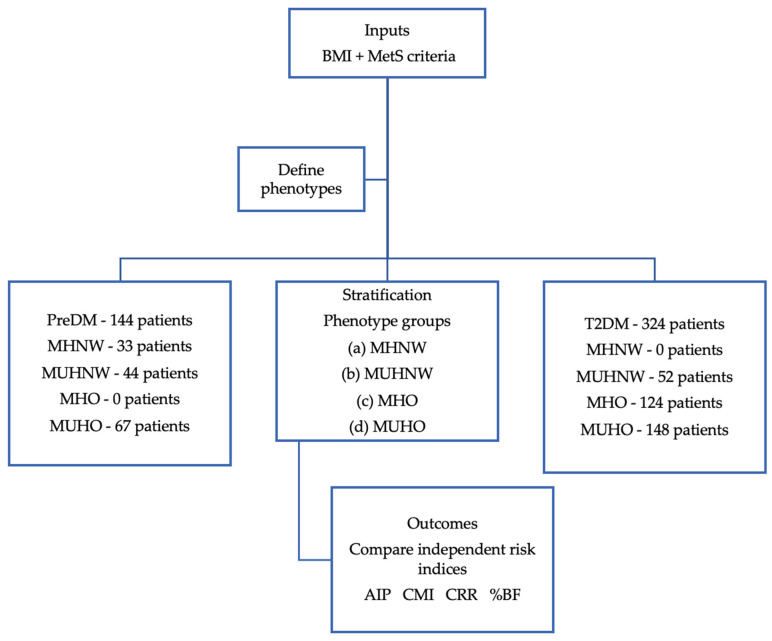
Conceptual framework of study design and analytical workflow. Distribution of the patients after the cardiometabolic phenotypes.

**Table 1 ijms-26-06227-t001:** Demographic and clinical characteristics, medical history, and laboratory assessments in the context of PreDM and T2DM.

Parameters	PreDM Cohort (*n* = 100)	T2DM Cohort (*n* = 200)	*p-Value from* *Pearson’s Chi-Squared/* *Student’s t-Test/* *Mann–Whitney Test*
** *Demographic and Clinical Features—Medical History* **			
Age (years) (mean ± SD)	56.51 ± 11.52	62.91 ± 12.32	<0.0001 *
Gender Male/Female (*n*)	47/53	101/99	0.56
Residence Urban/Rural (*n*)	61/39	119/81	0.75
Education Yes/No (*n*)	68/32	130/70	0.60
Drinking Yes/No (*n*)	42/58	60/140	0.03 *
Smoking Yes/No (*n*)	47/53	66/134	0.01 *
Hepatoseatosis *n* (%)	66 (66%)	148 (74%)	0.14
Dyslipidemia *n* (%)	85 (85%)	176 (88%)	0.46
Hypertension *n* (%)	93 (93%)	181 (90%)	0.46
Hyperuricemia *n* (%)	12 (12%)	35 (17%)	0.21
** *Laboratory examination* **			
HbA1c (%) (mean ± SD)	5.85 ± 0.36	9.36 ± 2.24	<0.0001 *
FPG (mg/dL) (mean ± SD)	99.47 ± 13.45	163.70 ± 41.00	<0.0001 *
2h-PG (mg/dL) (mean ± SD)	162.40 ± 13.91	259.90 ± 86.22	<0.0001 *
ESR (mm/1st hour) [median (range)]	30.00 (5.00–115.00)	30.00 (4.00–140.00)	0.02 *
CRP (mg/dL) [median (range)]	0.46 (0.01–30.42)	20.00 (3.20–98.80)	<0.0001 *
AST (mg/dL) [median (range)]	19.51 (9.85–75.08)	22.47 (9.85–146.20)	0.003 *
ALT (mg/dL) [median (range)]	19.50 (7.00–90.00)	23.00 (7.00–261.00)	0.0004 *
Creatinine (mg/dL) [median (range)]	0.77 (0.36–1.75)	0.78 (0.40–7.42)	0.90
e-GFR (mL/min/1.73 m^2^) MDRD equation (mean ± SD)	89.20 ± 33.67	88.34 ± 35.43	0.84
WBC (×10^3^/μL) (mean ± SD)	7.82 ± 2.10	8.83 ± 3.14	0.003 *
HGB (g/dL) (mean ± SD)	12.99 ± 2.06	13.26 ± 2.01	0.27
PLT (×10^3^/μL) (mean ± SD)	255.00 ± 75.48	254.80 ± 70.93	0.87

2h-PG: two-hour plasma glucose after a 75 g oral glucose tolerance test; AST: aspartate aminotransferase; ALT: alanine aminotransferase; MDRD: Modification of Diet in Renal Disease; CRP: C-reactive protein; e-GFR: estimated glomerular filtration rate; ESR: erythrocyte sedimentation rate; FPG: fasting plasma glucose; HGB: hemoglobin; HbA1c: glycosylated hemoglobin A1c; PLTs: platelets; SD: standard deviation; and WBC: white blood cells/leukocytes. * *p* < 0.05: statistically significant.

**Table 2 ijms-26-06227-t002:** Clinical features, anthropometric measurements, metabolic indices, lipid profiles, and metabolic syndrome-related parameters.

Parameters	PreDM Cohort (*n* = 100)	T2DM Cohort (*n* = 200)	*p-Value from* *Pearson’s Chi-Squared/* *Student’s t-Test/* *Mann–Whitney Test*
** *Clinical and Anthropometric Features* ** ** *—* ** ** *Various indices* **			
SBP (mmHg) (mean ± SD)	137.00 ± 17.91	136.3 ± 17.13	0.74
DBP (mmHg) (mean ± SD)	79.74 ± 14.41	80.86 ± 11.94	0.34
Height (cm) (mean ± SD)	166.3 ± 10.57	166.5 ± 9.70	0.66
Weight (kg) [median (range)]	82.00 (50.00–140.50)	80.00 (45.00–147.00)	0.77
WC (cm) [median (range)]	101.00 (58.00–171.00)	101.00 (52.00–133.00)	0.12
HC (cm) [median (range)]	108.50 (75.00–147.00)	108.00 (75.00–145.00)	0.93
WHR (mean ± SD)	0.93 (0.59–1.64)	0.92 (0.54–1.28)	0.052 **
WHtR [median (range)]	0.61 (0.34–0.95)	0.60 (0.35–0.85)	0.02 *
BMI (kg/m^2^) [median (range)]	30.40 (16.72–58.48)	29.55 (18.21–46.85)	0.80
%BF (mean ± SD)	39.10 ± 9.13	40.17 ± 9.35	0.34
** *Lipemic specter* **			
TC (mg/dL) (mean ± SD)	206.60 ± 58.78	187.5 ± 60.73	0.001 *
LDL-c (mg/dL) (mean ± SD)	125.10 ± 52.47	110.00 ± 47.71	0.005 *
HDL-c (mg/dL) (mean ± SD)	53.59 ± 13.59	44.12 ± 13.34	<0.0001 *
TG (mg/dL) [median (range)]	128.50 (45.00–610.00)	144.50 (44.00–1563.00)	0.34
**Metabolic Syndrome-Related Indices**			
TyG (mean ± SD)	8.78 ± 0.50	9.31 ± 0.67	<0.0001 *
TyG-BMI (mean ± SD)	268.10 ± 64.33	284.60 ± 59.59	0.02 *
TyG-WC (mean ± SD)	903.10 ± 137.10	910.10 ± 182.10	0.43
TyG-WHtR (mean ± SD)	5.43 ± 0.75	5.46 ± 1.06	0.77
TG/HDL-c [median (range)]	2.53 (0.59–19.06)	3.23 (0.64–45.97)	0.0005 *
AIP (mean ± SD) range	0.40 ± 0.26 (−0.23–1.28)	0.52 ± 0.30 (−0.19–1.66)	0.0005 *
CMI [median (range)]	1.59 (0.33–11.92)	1.89 (0.37–33.53)	0.008 *
CRR [median (range)]	3.71 (1.74–9.34)	4.06 (2.06–15.14)	0.02 *

SBP: systolic blood pressure; DBP: diastolic blood pressure; HDL-c: high-density lipoprotein cholesterol; LDL-c: low-density lipoprotein cholesterol; TC: total cholesterol; TG: total triglycerides; HC: hip circumference; WC: waist circumference; WHR: waist-to-hip ratio; WHtR: waist-to-height ratio; BMI: body mass index; %BF: body fat percentage; AIP: atherogenic index plasma; CMI: cardiometabolic index; CRR: cardiovascular risk ratio; and TyG: triglyceride-glucose index. * *p* < 0.05: statistically significant; **: stretched the significance limit.

**Table 3 ijms-26-06227-t003:** Comparing the cardiometabolic syndrome phenotypes in individuals with PreDM and T2DM.

Parameters	MUHNW-PreDM Cohort (*n* = 33)	MUHO-PreDM Cohort (*n* = 67)	*p*-Value from Pearson’s Chi-Squared/ Student’s *t*-Test/Mann–Whitney Test	MUHNW-T2DM Cohort (*n* = 52)	MUHO-T2DM Cohort (*n* = 148)	*p*-Value from Pearson’s Chi-Squared/ Student’s *t*-Test/Mann–Whitney Test
** *Demographic and Clinical Features—Medical History* **						
Age (years) (mean ± SD)	54.67 ± 11.46	57.42 ± 11.53	0.26	64.58 ± 13.41	62.32 ± 11.90	0.057 **
Gender Male/Female (*n*)	11/22	36/31	0.05 *	28/24	73/75	0.57
Residence Urban/Rural (*n*)	22/11	39/28	0.41	35/17	84/64	0.18
Education Yes/No (*n*)	26/7	42/25	0.10	40/12	90/58	0.03 *
Drinking Yes/No (*n*)	12/21	30/37	0.42	21/31	39/109	0.054 **
Smoking Yes/No (*n*)	14/19	33/34	0.52	19/33	47/101	0.52
Hepatoseatosis *n* (%)	26 (78%)	40 (59%)	0.05 *	27 (51%)	121 (81%)	<0.0001 *
Hyperuricemia *n* (%)	3 (9%)	9 (13%)	0.74	6 (11%)	29 (19%)	0.18
** *Anthropometric features and various indices* **						
Height (cm) (mean ± SD)	174.00 ± 10.70	162.60 ± 8.26	<0.0001 *	169.40 ± 9.48	165.50 ± 9.60	0.01 *
Weight (kg) [median (range)]	67.00 (50.00–94.00)	88.00 (65.00–140.5)	<0.0001 *	68.50 (45.00–87.00)	86.00 (62.00–147.00)	<0.0001 *
HC (cm) [median (range)]	104.00 (75.00–147.00)	111.00 (82.00–147.00)	0.11	103.50 (82.00–130.00)	110.00 (75.00–145.00)	<0.0001 *
WHtR [median (range)]	0.57 (0.34–0.95)	0.63 (0.45–0.88)	0.001 *	0.46 (0.35–0.70)	0.63 (0.43–0.85)	<0.0001 *
%BF (mean ± SD)	30.82 ± 6.77	42.90 ± 7.40	<0.0001 *	32.10 ± 7.70	42.68 ± 8.42	<0.0001 *
** *Laboratory examination* **						
ESR (mm/1st hour) [median (range)]	30.00 (10.00–115.00)	29.00 (5.00–115.00)	0.30	30.00 (4.00–110.00)	32.00 (4.00–140.00)	0.33
CRP (mg/dL) [median (range)]	19.50 (7.00–80.00)	20 (3.20–98.80)	0.71	0.30 (0.01–19.65)	0.51 (0.01–30.42)	0.01 *
AST (mg/dL) [median (range)]	19.35 (9.85–32.11)	19.62 (9.85–75.08)	0.41	21.28 (9.85–65.43)	22.64 (12.00–146.20)	0.05 *
ALT (mg/dL) [median (range)]	20.00 (7.00–28.00)	19.00 (7.00–90.00)	0.08	18.50 (7.00–60.00)	24.50 (9.00–261.00)	<0.0001 *
Creatinine (mg/dL) [median (range)]	0.74 (0.47–1.75)	0.82 (0.36–1.66)	0.61	0.71 (0.40–2.83)	0.82 (0.44–7.42)	0.15
e-GFR (mL/min/1.73 m^2^) MDRD equation (mean ± SD)	88.81 ± 37.25	89.39 ± 32.05	0.93	96.33 ± 41.93	85.53 ± 32.54	0.05 *
WBC (×10^3^/μL) (mean ± SD)	7.98 ± 2.28	7.73 ± 2.01	0.57	8.22 ± 2.77	9.04 ± 3.25	0.10
HGB (g/dL) (mean ± SD)	13.19 ± 1.86	12.89 ± 2.16	0.49	12.64 ± 1.87	13.48 ± 2.02	0.009 *
PLT (×10^3^/μL) (mean ± SD)	275.10 ± 89.42	245.10 ± 66.09	0.06	243.90 ± 78.36	258.70 ± 68.00	0.19
** *Metabolic Syndrome-Related Indices* **						
TyG-WHtR (mean ± SD)	5.04 ± 0.80	5.62 ± 0.66	0.0003 *	4.37 ± 0.68	5.84 ± 0.90	<0.0001 *
TG/HDL-c [median (range)]	2.35 (0.59–15.61)	2.54 (0.69–19.06)	0.91	2.55 (0.64–10.75)	3.47 (0.65–45.97)	0.001 *
AIP (mean ± SD) range	0.37 ± 0.31 (−0.23–1.19)	0.42 ± 0.24 (−0.16–1.28)	0.41	0.40 ± 0.26 (−0.19–1.03)	0.56 ± 0.30 (−0.19–1.66)	0.001 *
CMI [median (range)]	1.40 (0.33–8.98)	1.67 (0.45–11.92)	0.10	1.27 (0.36–4.89)	2.16 (0.43–33.53)	<0.0001 *
CRR [median (range)]	3.68 (1.78–8.68)	3.71 (1.74–9.34)	0.57	3.85 (2.06–7.42)	4.15 (2.33–15.14)	0.04 *

2h-PG: two-hour plasma glucose after a 75 g oral glucose tolerance test; AST: aspartate aminotransferase; ALT: alanine aminotransferase; MDRD: Modification of Diet in Renal Disease; CRP: C-reactive protein; e-GFR: estimated glomerular filtration rate; ESR: erythrocyte sedimentation rate; FPG: fasting plasma glucose; HGB: hemoglobin; HbA1c: glycosylated hemoglobin A1c; PLTs: platelets; SD: standard deviation; WBC: white blood cells/leukocytes; SBP: systolic blood pressure; DBP: diastolic blood pressure; HDL-c: high-density lipoprotein cholesterol; LDL-c: low-density lipoprotein cholesterol; TC: total cholesterol; TG: total triglycerides; HC: hip circumference; WC: waist circumference; WHR: waist-to-hip ratio; WHtR: waist-to-height ratio; BMI: body mass index; %BF: body-fat percentage; AIP: atherogenic index plasma; CMI: cardiometabolic index; CRR: cardiovascular risk ratio; TyG: triglyceride-glucose index; TyG-BMI: TyG–body mass index; TyG-WHtR: TyG–waist-to-height ratio; and TyG-WC: TyG–waist circumference. * *p* < 0.05: statistically significant; **: stretched the significance limit.

**Table 4 ijms-26-06227-t004:** Comparing the PreDM MUHNW patients with T2DM MUHNW patients and PreDM MUHO patients with T2DM MUHO patients.

Parameters	MUHNW-PreDM Cohort (*n* = 33)	MUHNW-T2DM Cohort (*n* = 52)	*p*-Value from Pearson’s Chi-Squared/ Student’s *t*-Test/Mann–Whitney Test	MUHO-PreDM Cohort (*n* = 67)	MUHO-T2DM Cohort (*n* = 148)	*p*-Value from Pearson’s Chi-Squared/ Student’s *t*-Test/Mann–Whitney Test
Age (years) (mean ± SD)	54.67 ± 11.46	64.58 ± 13.41	0.0007 *	57.42 ± 11.53	62.32 ± 11.90	0.005 *
Gender Male/Female (*n*)	11/22	28/24	0.06	36/31	73/75	0.54
Residence Urban/Rural (*n*)	22/11	35/17	1	39/28	84/64	0.84
Drinking Yes/No (*n*)	12/21	21/31	0.70	30/37	39/109	0.007 *
Smoking Yes/No (*n*)	14/19	19/33	0.59	33/34	47/101	0.01 *
Hepatoseatosis *n* (%)	26 (78%)	27 (51%)	0.01 *	40 (59%)	121 (81%)	0.0005 *
Height (cm) (mean ± SD)	174.00 ± 10.70	169.40 ± 9.48	0.04 *	162.60 ± 8.26	165.50 ± 9.60	0.03 *
Weight (kg) [median (range)]	67.00 (50.00–94.00)	68.50 (45.00–87.00)	0.65	88.00 (65.00–140.5)	86.00 (62.00–147.00)	0.54
HC (cm) [median (range)]	104.00 (75.00–147.00)	103.50 (82.00–130.00)	0.18	111.00 (82.00–147.00)	110.00 (75.00–145.00)	0.64
WHtR [median (range)]	0.57 (0.34–0.95)	0.46 (0.35–0.70)	<0.0001 *	0.63 (0.45–0.88)	0.63 (0.43–0.85)	0.28
%BF (mean ± SD)	30.82 ± 6.77	32.10 ± 7.70	0.43	42.90 ± 7.40	42.68 ± 8.42	0.85
e-GFR (mL/min/1.73 m^2^) MDRD equation (mean ± SD)	88.81 ± 37.25	96.33 ± 41.93	0.40	89.39 ± 32.05	85.53 ± 32.54	0.41
TyG-WHtR (mean ± SD)	5.04 ± 0.80	4.37 ± 0.68	<0.0001 *	5.62 ± 0.66	5.84 ± 0.90	0.059 **
TG/HDL-c [median (range)]	2.35 (0.59–15.61)	2.55 (0.64–10.75)	0.59	2.54 (0.69–19.06)	3.47 (0.65–45.97)	0.0002 *
AIP (mean ± SD) range	0.37 ± 0.31 (−0.23–1.19)	0.40 ± 0.26 (−0.19–1.03)	0.61	0.42 ± 0.24 (−0.16–1.28)	0.56 ± 0.30 (−0.19–1.66)	0.001 *
CMI [median (range)]	1.40 (0.33–8.98)	1.27 (0.36–4.89)	0.46	1.67 (0.45–11.92)	2.16 (0.43–33.53)	0.0008 *
CRR [median (range)]	3.68 (1.78–8.68)	3.85 (2.06–7.42)	0.91	3.71 (1.74–9.34)	4.15 (2.33–15.14)	0.01 *

2h-PG: two-hour plasma glucose after a 75 g oral glucose tolerance test; AST: aspartate aminotransferase; ALT: alanine aminotransferase; MDRD: Modification of Diet in Renal Disease; CRP: C-reactive protein; e-GFR: estimated glomerular filtration rate; ESR: erythrocyte sedimentation rate; FPG: fasting plasma glucose; HGB: hemoglobin; HbA1c: glycosylated hemoglobin A1c; PLTs: platelets; SD: standard deviation; WBC: white blood cells/leukocytes; SBP: systolic blood pressure; DBP: diastolic blood pressure; HDL-c: high-density lipoprotein cholesterol; LDL-c: low-density lipoprotein cholesterol; TC: total cholesterol; TG: total triglycerides; HC: hip circumference; WC: waist circumference; WHR: waist-to-hip ratio; WHtR: waist-to-height ratio; BMI: body mass index; %BF: body fat percentage; AIP: atherogenic index plasma; CMI: cardiometabolic index; CRR: cardiovascular risk ratio; TyG: triglyceride-glucose index; TyG-BMI: TyG–body mass index; TyG-WHtR: TyG–waist-to-height ratio; and TyG-WC: TyG–waist circumference. * *p* < 0.05: statistically significant; **: stretched the significance limit.

**Table 5 ijms-26-06227-t005:** Associations among TyG, TyG-related indices and BMI, WHR, WHtR, and %BF in patients with PreDM.

Variables (Mean ± SD)	PreDM Cohort (*n* = 100)							
	TyG	*p*-Value from Kruskal–Wallis/ One-Way ANOVA	TyG-BMI	*p*-Value from Kruskal–Wallis/ One-Way ANOVA	TyG-WC	*p*-Value from Kruskal–Wallis/ One-Way ANOVA	TyG-WHtR	*p*-Value from Kruskal–Wallis/ One-Way ANOVA
BMI category (kg/m^2^)								
Normal weight (18.5–24.9 kg/m^2^)	8.57 ± 0.55	0.05 *	185.20 ± 22.08	<0.0001 *	884.50 ± 196.40	0.47	5.01 ± 0.96	0.002 *
Overweight (25–29.9 kg/m^2^)	8.92 ± 0.53		242.60 ± 21.38		886.60 ± 107.80		5.32 ± 0.57	
Obese (≥30 kg/m^2^)	8.80 ± 0.45		313.30 ± 49.42		919.20 ± 119.70		5.65 ± 0.66	
WHR								
Q 1 (0.59–0.86)	8.65 ± 0.42	0.17	286.40 ± 77.51	0.25	785.00 ± 122.30	<0.0001 *	4.90 ± 0.78	<0.0001 *
Q 2 (0.87–0.92)	8.80 ± 0.40		260.00 ± 53.20		919.00 ± 118.90		5.57 ± 0.62	
Q 3 (0.93–0.97)	8.96 ± 0.59		250.10 ± 50.78		905.90 ± 99.05		5.34 ± 0.57	
Q 4 (0.98–1.64)	8.73 ± 0.55		268.70 ± 70.74		990.50 ± 119.80		5.85 ± 0.70	
WHtR								
Q 1 (0.34–0.56)	8.86 ± 0.61	0.51	242.90 ± 78.24	0.09	775.40 ± 130.00	<0.0001 *	4.59 ± 0.64	<0.0001 *
Q 2 (0.57–0.60)	8.86 ± 0.42		261.60 ± 53.64		880.20 ± 73.76		5.25 ± 0.29	
Q 3 (0.61–0.66)	8.71 ± 0.54		272.70 ± 46.13		914.00 ± 80.58		5.55 ± 0.35	
Q 4 (0.67–0.95)	8.70 ± 0.40		289.70 ± 77.33		1051.00 ± 113.30		6.34 ± 0.50	
%BF								
Q 1 (20.49–32.58)	8.72 ± 0.59	0.22	201.10 ± 38.88	<0.0001 *	894.80 ± 173.80	0.04 *	5.16 ± 0.84	0.004 *
Q 2 (32.59–39.04)	8.64 ± 0.41		254.80 ± 32.06		848.90 ± 91.73		5.26 ± 0.58	
Q 3 (39.05–45.54)	8.89 ± 0.43		277.60 ± 46.77		913.50 ± 114.90		5.43 ± 0.62	
Q 4 (45.55–67.32)	8.88 ± 0.55		333.40 ± 58.28		955.10 ± 140.10		5.86 ± 0.79	

WHR: waist-to-hip ratio; WHtR: waist-to-height ratio; BMI: body mass index; %BF: body fat percentage; and TyG: triglyceride-glucose index. * *p* < 0.05: statistically significant.

**Table 6 ijms-26-06227-t006:** Associations among TyG, TyG-related indices and BMI, WHR, WHtR, and %BF in patients with T2DM.

Variables (Mean ± SD)	T2DM Cohort (*n* = 200)							
	TyG	*p*-Value from Kruskal–Wallis/ One-Way ANOVA	TyG-BMI	*p*-Value from Kruskal–Wallis/ One-Way ANOVA	TyG-WC	*p*-Value from Kruskal–Wallis/ One-Way ANOVA	TyG-WHtR	*p*-Value from Kruskal–Wallis/ One-Way ANOVA
BMI category (kg/m^2^)								
Normal weight (18.5–24.9 kg/m^2^)	9.06 ± 0.56	0.02 *	212.10 ± 18.53	<0.0001 *	724.30 ± 132.40	<0.0001 *	4.26 ± 0.60	<0.0001 *
Overweight (25–29.9 kg/m^2^)	9.37 ± 0.63		259.30 ± 24.71		880.70 ± 152.30		5.25 ± 0.79	
Obese (≥30 kg/m^2^)	9.39 ± 0.72		328.10 ± 51.19		1009.00 ± 145.70		6.11 ± 0.83	
WHR								
Q 1 (0.54–0.79)	9.27 ± 0.70	0.88	240.60 ± 48.19	<0.0001 *	680.60 ± 105.90	<0.0001 *	4.18 ± 0.56	<0.0001 *
Q 2 (0.80–0.91)	9.34 ± 0.69		291.50 ± 59.88		916.30 ± 122.00		5.41 ± 0.70	
Q 3 (0.92–0.97)	9.28 ± 0.54		298.20 ± 54.85		979.70 ± 109.10		5.84 ± 0.64	
Q 4 (0.98–1.28)	9.36 ± 0.73		300.00 ± 61.56		1060.00 ± 124.00		6.39 ± 0.82	
WHtR								
Q 1 (0.35–0.50)	9.24 ± 0.60	0.54	229.40 ± 33.32	<0.0001 *	672.60 ± 80.79	<0.0001 *	4.12 ± 0.42	<0.0001 *
Q 2 (0.51–0.59)	9.30 ± 0.76		266.20 ± 43.21		897.50 ± 96.82		5.24 ± 0.49	
Q 3 (0.60–0.65)	9.29 ± 0.59		294.50 ± 42.18		983.30 ± 86.65		5.84 ± 0.37	
Q 4 (0.66–0.85)	9.43 ± 0.74		344.00 ± 59.91		1097.00 ± 116.80		6.71 ± 0.70	
%BF								
Q 1 (13.27–33.11)	9.19 ± 0.72	0.15	235.10 ± 40.51	<0.0001 *	859.30 ± 172.50	<0.0001 *	4.96 ± 0.91	<0.0001 *
Q 2 (33.12–40.41)	9.36 ± 0.57		272.60 ± 38.09		894.90 ± 170.40		5.28 ± 0.86	
Q 3 (40.42–46.29)	9.23 ± 0.70		283.00 ± 50.31		877.40 ± 200.50		5.33 ± 1.09	
Q 4 (46.30–63.23)	9.46 ± 0.66		343.70 ± 57.67		1009.00 ± 151.90		6.30 ± 0.91	

WHR: waist-to-hip ratio; WHtR: waist-to-height ratio; BMI: body mass index; %BF: body fat percentage; TyG: triglyceride-glucose index; TyG-BMI: TyG–body mass index; TyG-WHtR: TyG–waist-to-height ratio; and TyG-WC: TyG–waist circumference. * *p* < 0.05: statistically significant.

**Table 7 ijms-26-06227-t007:** Associations among AIP, CMI, CRR and BMI, WHR, WHtR, and %BF in patients with PreDM and T2DM.

	Variables (Mean ± SD) [Median (Range)]	PreDM Cohort (*n* = 100)						T2DM Cohort (*n* = 200)					
		AIP	*p*-Value from Kruskal–Wallis/ One-Way ANOVA	CMI	*p*-Value from Kruskal–Wallis/ One-Way ANOVA	CRR	*p*-Value from Kruskal–Wallis/ One-Way ANOVA	AIP	*p*-Value from Kruskal–Wallis/ One-Way ANOVA	CMI	*p*-Value from Kruskal–Wallis/ One-Way ANOVA	CRR	*p*-Value from Kruskal–Wallis/ One-Way ANOVA
	BMI category (kg/m^2^)												
	Normal weight (18.5–24.9 kg/m^2^)	0.30 ± 0.30 (−0.23–1.19)	0.06	1.26 (0.33–8.89)	0.01 *	3.60 (2.07–8.68)	0.49	0.37 ± 0.23 (−0.03–0.98)	0.001 *	1.09 (0.47–4.13)	<0.0001 *	3.50 (2.06–7.42)	0.01 *
	Overweight (25–29.9 kg/m^2^)	0.47 ± 0.25 (0.02–1.05)		1.85 (0.59–6.26)		4.09 (1.78–8.26)		0.55 ± 0.27 (−0.19–1.06)		2.02 (0.36–7.08)		4.38 (2.10–10.17)	
	Obese (≥30 kg/m^2^)	0.41 ± 0.24 (−0.16–1.28)		1.65 (0.45–11.92)		3.61 (1.74–9.34)		0.56 ± 0.32 (−0.19–1.66)		2.21 (0.43–33.53)		4.16 (2.36–15.14)	
	WHR												
PreDM Value T2DM Value	Q 1 (0.59–0.86) (0.54–0.79)	0.31 ± 0.21 (−0.23–0.75)	0.04 *	1.23 (0.33–3.38)	0.02 *	3.19 (1.74–6.64)	0.19	0.49 ± 0.33 (−0.06–1.47)	0.001 *	1.28 (0.47–15.43)	0.03 *	4.02 (2.06–8.14)	0.48
PreDM Value T2DM Value	Q 2 (0.87–0.92) (0.80–0.91)	0.39 ± 0.19 (−0.03–0.89)		1.62 (0.58–5.53)		3.52 (1.78–8.26)		0.53 ± 0.29 (−0.19 ± 1.29)		1.99 (0.43–11.24)		4.46 (2.33–15.14)	
PreDM Value T2DM Value	Q 3 (0.93–0.97) (0.92–0.97)	0.52 ± 0.30 (−0.13–1.19)		1.88 (0.41–8.89)		3.93 (2.08–6.06)		1.04 ± 1.52 (−0.19–8.81)		1.95 (0.36–6.66)		3.91 (2.10–10.17)	
PreDM Value T2DM Value	Q 4 (0.98–1.64) (0.98–1.28)	0.39 ± 0.29 (−0.22–1.28)		1.73 (0.39–11.92)		3.95 (2.07–9.34)		0.54 ± 0.33 (−0.03–1.66)		2.21 (0.59–33.53)		3.90 (2.33–10.44)	
	WHtR												
PreDM Value T2DM Value	Q 1 (0.34–0.56) (0.35–0.50)	0.42 ± 0.30 (−0.23–1.19)	0.92	1.39 (0.33–8.89)	0.47	3.62 (2.72–8.26)	0.57	0.48 ± 0.27 (0.04–1.03)	0.41	1.35 (0.47–4.89)	<0.0001 *	4.12 (2.06–8.14)	0.60
PreDM Value T2DM Value	Q 2 (0.57–0.60) (0.51–0.59)	0.42 ± 0.17 (0.02–0.78)		1.62 (0.59–3.54)		3.52 (2.08–8.24)		0.49 ± 0.32 (−0.19–1.47)		1.82 (0.36–15.43)		4.09 (2.10–15.14)	
PreDM Value T2DM Value	Q 3 (0.61–0.66) (0.60–0.65)	0.39 ± 0.33 (−0.22–1.28)		1.53 (0.39–11.92)		3.84 (1.78–9.34)		0.53 ± 0.27 (−0.03–1.18)		2.10 (0.59–9.42)		3.80 (2.46–13.17)	
PreDM Value T2DM Value	Q 4 (0.67–0.95) (0.66–0.85)	0.38 ± 0.20 (0.07–0.91)		1.73 (0.82–5.99)		3.77 (1.74–8.68)		0.57 ± 0.33 (−0.19–1.66)		2.50 (0.43–33.53)		4.22 (2.33–10.44)	
	%BF												
PreDM Value T2DM Value	Q 1 (20.49–32.58) (13.27–33.11)	0.38 ± 0.32 (−0.23 ± 1.19)	0.48	1.44 (0.33–8.89)	0.27	4.29 (2.07–8.68)	0.34	0.46 ± 0.32 (−0.19–1.47)	0.53	1.58 (0.36–15.43)	0.06	4.06 (2.10–7.42)	0.68
PreDM Value T2DM Value	Q 2 (32.59–39.04) (33.12–40.41)	0.35 ± 0.22 (−0.03–1.05)		1.41 (0.58–6.26)		3.63 (1.78–7.29)		0.54 ± 0.27 (0.03–1.18)		2.02 (0.48–9.42)		3.92 (2.06–13.17)	
PreDM Value T2DM Value	Q 3 (39.05–45.54) (40.42–46.29)	0.45 ± 0.20 (0.17–0.89)		1.70 (0.73–5.53)		3.93 (2.08–6.53)		0.53 ± 0.30 (−0.09–1.66)		1.82 (0.47–33.53)		4.32 (2.42–10.44)	
PreDM Value T2DM Value	Q 4 (45.55–67.32) (46.30–63.23)	0.43 ± 0.30 (−0.16–1.28)		1.73 (0.45–11.92)		3.37 (1.74–9.34)		0.54 ± 0.31 (−0.19 ± 1.29)		2.17 (0.43–11.92)		3.92 (2.37–15.14)	

WHR: waist-to-hip ratio; WHtR: waist-to-height ratio; BMI: body mass index; %BF: body fat percentage; AIP: atherogenic index plasma; CMI: cardiometabolic index; and CRR: cardiovascular risk ratio. * *p* < 0.05: statistically significant.

**Table 8 ijms-26-06227-t008:** Comparing the results of the MetS-related indices between males and females.

Metabolic Syndrome- Related Indices	Male—PreDM (*n* = 47)	Male—T2DM (*n* = 101)	*p*-Value from Pearson’s Chi-Squared/ Student’s *t*-Test	Female—PreDM (*n* = 53)	Female—T2DM (*n* = 99)	*p*-Value from Pearson’s Chi-Squared/ Student’s *t*-Test
TyG (mean ± SD)	8.70 ± 0.40	9.26 ± 0.64	<0.0001 *	8.85 ± 0.57	9.37 ± 0.69	<0.0001 *
TyG-BMI (mean ± SD)	279.7 ± 70.10	272.80 ± 54.60	0.51	255.30 ± 58.77	292.90 ± 65.96	0.0007 *
TyG-WC (mean ± SD)	886.20 ± 136.50	937.70 ± 153.00	0.05 *	918.10 ± 137.10	882.00 ± 204.60	0.25
TyG-WHtR (mean ± SD)	5.46 ± 0.74	5.44 ± 0.84	0.87	5.40 ± 0.77	5.49 ± 1.25	0.64
TG/HDL-c [median (range)]	2.42 (0.60–9.59)	3.41 (0.64–29.35)	0.001 *	2.58 (0.59–19.06)	3.19 (0.65–45.97)	0.07
AIP (mean ± SD) range	0.37 ± 0.21 (−0.22–0.98)	0.51 ± 0.29 (−0.19–1.47)	0.004 *	0.43 ± 0.30 (−0.23–1.28)	0.52 ± 0.31 (−0.19–1.66)	0.08
CMI [median (range)]	1.60 (0.39–6.16)	2.02 (0.36–15.43)	0.009 *	1.58 (0.33–11.92)	1.82 (0.43–33.53)	0.27
CRR [median (range)]	3.91 (2.07–8.68)	4.05 (2.10–13.17)	0.63	3.52 (1.74–9.34)	4.14 (2.06–15.14)	0.01 *
%BF (mean ± SD)	35.69 ± 9.31	33.54 ± 6.65	0.11	41.77 ± 8.08	46.45 ± 7.14	0.0003 *

%BF: body fat percentage; AIP: atherogenic index plasma; CMI: cardiometabolic index; CRR: cardiovascular risk ratio; TyG: triglyceride-glucose index; TyG-BMI: TyG–body mass index; TyG-WHtR: TyG–waist-to-height ratio; and TyG-WC: TyG–waist circumference. * *p* < 0.05: statistically significant.

**Table 9 ijms-26-06227-t009:** Diagnostic performance of the investigated parameters.

Parameter	AUC	Std. Error	Cut-Off Values	Sensitivity (%)	Specificity (%)	Youden Index	*p*-Value
** *PreDM-T2DM* **							
AIP	0.623	0.033	0.49	54.00	67.00	0.21	0.0005 *
TG/HDLc	0.622	0.033	2.79	62.50	61.00	0.24	0.0005 *
CMI	0.592	0.033	1.71	54.00	59.00	0.13	0.009 *
WHtR	0.579	0.033	0.60	50.50	57.00	0.08	0.02 *
CRR	0.577	0.035	3.79	53.00	59.00	0.12	0.02 *
%BF	0.527	0.035	40.05	53.50	54.00	0.08	0.435
TyG-WHtR	0.517	0.033	5.56	50.00	64.00	0.14	0.628
** *MUHNW-PreDM—MUHNW-T2DM* **							
WHtR	0.798	0.050	0.52	71.15	84.85	0.56	<0.0001 *
TyG-WHtR	0.765	0.054	4.78	73.08	69.70	0.43	<0.0001 *
CRR	0.549	0.063	3.46	42.31	63.64	0.06	0.443
CMI	0.547	0.064	1.39	57.69	51.52	0.09	0.462
TG/HDLc	0.534	0.064	2.37	55.77	51.52	0.07	0.594
AIP	0.533	0.064	0.37	55.77	51.52	0.07	0.604
%BF	0.506	0.065	29.14	57.69	48.48	0.06	0.917
** *MUHO-PreDM—MUHO-T2DM* **							
TG/HDLc	0.659	0.038	2.92	63.51	68.66	0.32	0.0002 *
AIP	0.659	0.038	0.44	67.57	64.18	0.32	0.0002 *
CMI	0.641	0.039	1.79	65.54	61.19	0.27	0.0009 *
%BF	0.607	0.042	42.55	51.53	50.75	0.02	0.01 *
CRR	0.591	0.043	3.87	64.19	58.21	0.22	0.03 *
TyG-WHtR	0.587	0.040	5.61	61.49	58.21	0.19	0.03 *
WHtR	0.545	0.041	0.63	56.76	47.76	0.05	0.287

WHtR: waist-to-height ratio; BMI: body mass index; %BF: body fat percentage; AIP: atherogenic index plasma; CMI: cardiometabolic index; CRR: cardiovascular risk ratio; and TyG-WHtR: TyG–waist-to-height ratio. * *p* < 0.05: statistically significant.

**Table 10 ijms-26-06227-t010:** Definition of the MetS criteria.

*Criteria*	*Cut-Off Values*
Waist circumference	>102 cm (male); >88 cm (female)
Serum FPG	≥100 mg/dL or use of hypoglycemic drugs
Serum tryglicerides	≥150 mg/dL or use of triglyceride-lowering drugs
Reduced HDL-c values	≤40 mg/dL (males) or ≤50 mg/dL (females), or use of HDL-C-raising drugs
Elevated blood pressure	SBP ≥ 130 mmHg or DBP ≥ 85 mmHg, or use of blood pressure-lowering drugs

**Table 11 ijms-26-06227-t011:** Definition of cardiometabolic phenotypes.

*Cardiometabolic Phenotype*	*Criteria*			
	BMI < 25.0 kg/m^2^	BMI ≥ 25.0 kg/m^2^	<3 MetS Criteria	≥3 MetS Criteria
MHNW	+	−	+	−
MUHNW	−	+	−	+
MHO	+	−	+	−
MUHO	−	+	−	+

+: present criteria; −: absent criteria; BMI: body mass index; refer to Table 10 for MetS criteria; MHNW: metabolically healthy normal weight; MUHNW: metabolically unhealthy normal weight; MHO: metabolically healthy obese; and MUHO: metabolically unhealthy obese.

## Data Availability

The data used to support the findings of this study are available from the corresponding author upon reasonable request. The data are not publicly available due to ethical reasons.

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
