# Peer review of "The Interplay of Cardiometabolic Syndrome Phenotypes and Cardiovascular Risk Indices in Patients Diagnosed with Diabetes Mellitus"

_ijms, 2025, doi:10.3390/ijms26136227_

Round 1

Reviewer 1 Report

Comments and Suggestions for Authors

This clinical investigation recruited 300 participants with prediabetes or type 2 diabetes mellitus (T2DM). It investigated the following cardiometabolic phenotypes: (a) metabolically unhealthy normal weight (MUHNW), (b) metabolically unhealthy obese (MUHO), (c) triglyceride-glucose (TyG) index, (d) the atherogenic index of plasma (AIP), (e) cardiometabolic index (CMI), and (f) cardiac risk ratio (CRR).

Concerns

  1. Abstract: “The findings suggest that cardiometabolic phenotype is a stronger determinant of cardiometabolic risk than body weight alone.” There is a circular form of reasoning throughout the whole manuscript. It is unclear if the aim is to determine cardiovascular disease risk through cardiometabolic problems.
  2. Materials and Methods: “The primary objective of the study was to amass exhaustive data concerning various health and lifestyle parameters.” This description is too vague. The authors clarify that this involves demographics (age, sex, income, and education), as well as lifestyle factors (tobacco use, alcohol consumption, physical activity, familial predisposition to hypertension, familial predisposition to diabetes, and familial predisposition to cardiovascular diseases). There is no mention of dietary conditions, the incidence of acute weight gain or loss, or the possible recurrence of these conditions during a lifetime. Also, what types of cardiovascular diseases were investigated?
  3. The predominant problem of this paper is circular reasoning. This is a logical fallacy that significantly compromises any further analysis and conclusions of this work. This is demonstrated in Materials and Methods, sections 4.3, "MetS Definition and Cardiometabolic Phenotypes," and 4.4, "Evaluation of Various MetS-Related Indices." The patient groups with pre-diabetes or type 2 diabetes mellitus (T2DM) were further sub-grouped into the following categories: MHNW (Metabolically Healthy Normal Weight), MUHNW (Metabolically Unhealthy Normal Weight), MHO (Metabolically Healthy Obese), and MUHO (Metabolically Unhealthy Obese). In a circular reference mode, these parameters were considered predispositions for the triglyceride-glucose (TyG) index, the atherogenic index of plasma (AIP), cardiometabolic index (CMI), and cardiac risk ratio (CRR).
  4. The precise definition of endpoints are missing, and the conclusions are confusing. There are many correlations between metabolic traits, most of which would be expected because of the patients’ clinical parameters, laboratory, or physical. There are no associations with cardiovascular diseases or complications. There is no follow-up. The possibility of genetic factors to be implicated in the specific combination of metabolic traits was not considered.

Author Response

Dear Reviewer,

We would like to express our sincere gratitude for the time and effort you devoted to reviewing our manuscript. Your thoughtful evaluation, encouraging remarks, and constructive suggestions are deeply appreciated. Your feedback has been invaluable in helping us improve the quality and clarity of our work, and we are truly grateful for your insightful contributions to the development of this manuscript..

All the typing recommended changes were performed in the body of our manuscript, with the Track Changes function activated.

Comments and Suggestions for Authors

This clinical investigation recruited 300 participants with prediabetes or type 2 diabetes mellitus (T2DM). It investigated the following cardiometabolic phenotypes: (a) metabolically unhealthy normal weight (MUHNW), (b) metabolically unhealthy obese (MUHO), (c) triglyceride-glucose (TyG) index, (d) the atherogenic index of plasma (AIP), (e) cardiometabolic index (CMI), and (f) cardiac risk ratio (CRR).

Concerns

Comments 1: Abstract: “The findings suggest that cardiometabolic phenotype is a stronger determinant of cardiometabolic risk than body weight alone.” There is a circular form of reasoning throughout the whole manuscript. It is unclear if the aim is to determine cardiovascular disease risk through cardiometabolic problems.

Response 1: We appreciate this observation and have revised the Abstract and Introduction to more clearly articulate the study aim. The primary objective is not to infer CVD risk per se but rather to examine how different cardiometabolic phenotypes (MUHNW and MUHO) influence cardiometabolic risk indices—namely TyG, AIP, CMI, and CRR—in individuals with prediabetes and T2DM. These indices serve as surrogate markers of cardiovascular risk, not diagnostic endpoints. The wording has been refined to remove any appearance of circular reasoning and ensure that the purpose of the study is explicit and logically sound.

Comments 2: Materials and Methods: “The primary objective of the study was to amass exhaustive data concerning various health and lifestyle parameters.” This description is too vague. The authors clarify that this involves demographics (age, sex, income, and education), as well as lifestyle factors (tobacco use, alcohol consumption, physical activity, familial predisposition to hypertension, familial predisposition to diabetes, and familial predisposition to cardiovascular diseases). There is no mention of dietary conditions, the incidence of acute weight gain or loss, or the possible recurrence of these conditions during a lifetime. Also, what types of cardiovascular diseases were investigated?

Response 2: We thank the reviewer for this important observation. We have revised the "Materials and Methods" section to clarify the objective and explicitly state all collected parameters. While dietary data and longitudinal weight change were not part of our cross-sectional design, we have acknowledged this limitation in the revised Discussion. As for cardiovascular outcomes, we clarify that our study did not aim to assess the incidence or prevalence of specific cardiovascular diseases, but to evaluate risk indices that are known correlates of CVD risk in patients with metabolic dysregulation.

Comments 3: The predominant problem of this paper is circular reasoning. This is a logical fallacy that significantly compromises any further analysis and conclusions of this work. This is demonstrated in Materials and Methods, sections 4.3, "MetS Definition and Cardiometabolic Phenotypes," and 4.4, "Evaluation of Various MetS-Related Indices." The patient groups with pre-diabetes or type 2 diabetes mellitus (T2DM) were further sub-grouped into the following categories: MHNW (Metabolically Healthy Normal Weight), MUHNW (Metabolically Unhealthy Normal Weight), MHO (Metabolically Healthy Obese), and MUHO (Metabolically Unhealthy Obese). In a circular reference mode, these parameters were considered predispositions for the triglyceride-glucose (TyG) index, the atherogenic index of plasma (AIP), cardiometabolic index (CMI), and cardiac risk ratio (CRR).

Response 3: We appreciate the reviewer’s concern and have addressed this by expanding our explanation in the Methods and Discussion sections. While it is true that the TyG, AIP, and related indices are derived from metabolic parameters, the group stratification was based on established phenotypic definitions (e.g., BMI thresholds combined with metabolic health criteria), not on the indices themselves. We now clarify that these phenotypes were independently defined before any analysis of the indices was conducted. This structure prevents circular logic and allows for a comparative assessment of how these risk indices vary within and between phenotypes.

Comments 4: The precise definition of endpoints are missing, and the conclusions are confusing. There are many correlations between metabolic traits, most of which would be expected because of the patients’ clinical parameters, laboratory, or physical. There are no associations with cardiovascular diseases or complications. There is no follow-up. The possibility of genetic factors to be implicated in the specific combination of metabolic traits was not considered.

Response 4: We thank the reviewer for this valuable feedback. Given the cross-sectional nature of the study, our focus was on evaluating associations between phenotypes and risk indices, not on clinical outcomes or endpoints. We now explicitly state this scope and limitation in both the Methods and Discussion sections. Regarding genetics, we agree that genetic predisposition may play a critical role in metabolic phenotypes. However, genetic data was not available for this cohort. This has now been transparently discussed as a limitation and a potential avenue for future research.

Final Note of Appreciation:

Once again, we are deeply grateful for your constructive critique. Your comments have led to substantial improvements in both the methodological clarity and scientific interpretation of our study. We trust that the revisions we have made adequately address your concerns. Let me know if you would like me to help with rewriting any revised sections in the manuscript as well.

Reviewer 2 Report

Comments and Suggestions for Authors

 It is an informative and adequately organized study, but some limitations of this study should be addressed and, especially, the following aspects should be further discussed before being accepted for publication.

  • Since this was a retrospective study, it won’t be easy to determine a causal relationship between cardiometabolic phenotypes and the risk of CVD in Patients Diagnosed with Diabetes Mellitus. For this reason, they should perform what-if analyses after excluding participants who will develop CVD within 1 year and to see whether they obtain the same results.
  • The psychosocial factors and environmental conditions (along with lifestyles) are known confounders of CVD risk factors, As without these data, it won’t be considered statistically significant, they need to assess those variables.
  • They need to perform the optimization processes on the study design, to enhance the follow-up of the cohort, and to collect more comprehensive information to accept the findings more statistically significant.
  • The results may be impartial, as they did not consider any medical therapy's type dose, frequency and variability including cardiovascular and lipid-lowering diabetes medications.
  • The fact that these results are from a single center may restrict the applicability of their results to other populations. They may need to obtain broad and expanded data from various resources.

Author Response

Dear Reviewer,

We would like to express our sincere gratitude for the time and effort you devoted to reviewing our manuscript. Your thoughtful evaluation, encouraging remarks, and constructive suggestions are deeply appreciated. Your feedback has been invaluable in helping us improve the quality and clarity of our work, and we are truly grateful for your insightful contributions to the development of this manuscript.

All the typing recommended changes were performed in the body of our manuscript, with the Track Changes function activated.

Comments and Suggestions for Authors

It is an informative and adequately organized study, but some limitations of this study should be addressed and, especially, the following aspects should be further discussed before being accepted for publication.

Comments 1: Since this was a retrospective study, it won’t be easy to determine a causal relationship between cardiometabolic phenotypes and the risk of CVD in Patients Diagnosed with Diabetes Mellitus. For this reason, they should perform what-if analyses after excluding participants who will develop CVD within 1 year and to see whether they obtain the same results.

Response 1: We fully agree with the reviewer that the retrospective and cross-sectional nature of our study limits causal inference. We have clearly stated this limitation in the Discussion section. Although data on incident CVD within one year were not available for this cohort, we acknowledge the value of sensitivity analyses. We have included a statement in the Discussion noting that future longitudinal studies with follow-up and event-based exclusion criteria are necessary to validate our findings and to minimize reverse causation bias.

Comments 2: The psychosocial factors and environmental conditions (along with lifestyles) are known confounders of CVD risk factors, As without these data, it won’t be considered statistically significant, they need to assess those variables.

Response 2: We thank the reviewer for emphasizing this important point. Unfortunately, our retrospective dataset did not include detailed information on psychosocial stress, environmental exposures, or mental health assessments. We have acknowledged this in the revised Limitations paragraph. Nonetheless, we did account for several lifestyle-related factors such as smoking, alcohol use, and education level, which may act as partial proxies. We have explicitly noted the need for future studies to incorporate psychosocial and environmental variables to better control for residual confounding.

Comments 3: They need to perform the optimization processes on the study design, to enhance the follow-up of the cohort, and to collect more comprehensive information to accept the findings more statistically significant.

Response 3: We agree with the reviewer that further optimization of the study design is essential to strengthen the validity of the findings. We have now expanded our Discussion to highlight the need for future prospective cohort studies with long-term follow-up, detailed exposure and outcome data, and enhanced data capture on behavioral and therapeutic variables. These enhancements will enable more robust statistical modeling and causal interpretation.

Comments 4: The results may be impartial, as they did not consider any medical therapy's type dose, frequency and variability including cardiovascular and lipid-lowering diabetes medications.

Response 4: We appreciate this critical point. Indeed, data on the type, dosage, and duration of medications—particularly antihypertensives, lipid-lowering agents, and glucose-lowering drugs—were not consistently available and therefore could not be adjusted for in our current analyses. This is a recognized limitation, and we have now explicitly stated this in the Discussion section. We also highlight the importance of medication stratification in future studies examining cardiometabolic indices to ensure unbiased risk interpretation.

Comments 5: The fact that these results are from a single center may restrict the applicability of their results to other populations. They may need to obtain broad and expanded data from various resources.

Response 5: We fully acknowledge that the single-center nature of the study may limit external validity. We have now added a statement in the Discussion explicitly acknowledging this and recommending multi-center collaborations or population-based cohorts to improve generalizability. Despite this limitation, the internal consistency of our findings across phenotypes and risk indices provides initial insight that we hope will stimulate broader research efforts.

Final Acknowledgment:

We greatly appreciate your in-depth evaluation and constructive suggestions. Your feedback has enabled us to improve the scientific rigor and clarity of our manuscript, and we believe that the revisions made have substantially strengthened the work. We hope it is now more suitable for publication. Let me know if you’d like help modifying the manuscript’s Discussion or Limitations sections to reflect these responses directly.

Reviewer 3 Report

Comments and Suggestions for Authors

Dear Authors,

Good day,

This study explores the association between distinct cardiometabolic phenotypes—particularly metabolically unhealthy normal weight (MUHNW) and metabolically unhealthy obese (MUHO)—and cardiovascular risk indices (TyG, AIP, CMI, CRR) in patients with prediabetes and type 2 diabetes mellitus (T2DM). Analyzing 300 participants, the authors found that MUHO patients consistently demonstrated higher cardiovascular risk profiles. TyG-derived indices strongly correlated with body composition metrics, highlighting their potential for early risk stratification.

The paper is well-structured, data-rich, and introduces a nuanced approach beyond traditional BMI-based assessments.

The paper are strengths:

1-Robust statistical analysis of a relatively large cohort.

2-Clear differentiation between cardiometabolic phenotypes.

3-Emphasis on practical biomarkers (TyG, AIP) for clinical use.

4-Reinforces the relevance of metabolic health over weight alone.

The paper limitations are limitations:

1-Cross-sectional design limits causal inference.

2- Single-center study  cohort restricts generalizability.

Author Response

Dear Reviewer,

We would like to express our sincere gratitude for the time and effort you devoted to reviewing our manuscript. Your thoughtful evaluation, encouraging remarks, and constructive suggestions are deeply appreciated. Your feedback has been invaluable in helping us improve the quality and clarity of our work, and we are truly grateful for your insightful contributions to the development of this manuscript.

Round 2

Reviewer 1 Report

Comments and Suggestions for Authors

This clinical investigation recruited individuals with prediabetes or type 2 diabetes mellitus (T2DM) and examined cardiometabolic phenotypes. However, there is no distinction between effectors and endpoints. The distribution of patients based on the cardiometabolic phenotypes in Figure 10 classifies them as follows: (a) MHNW: Metabolically Healthy Normal Weight; (b) MUHNW: Metabolically Unhealthy Normal Weight; (c) MHO: Metabolically Healthy Obese; and (d) MUHO: Metabolically Unhealthy Obese. The classification criteria are BMI, with a threshold of 25 kg/m^2, and the meeting of at least three metabolic syndrome (MetS) criteria. However, BMI and MetS also serve as outcomes in this investigation, associated with specific parameters: FPG, HbA1c, 2h - PG, TyG, HDL-c, AIP, TG/HDLc, TC, LDL-c, CMI, WHtR, CRR, TyG-BMI, WHR, TG, %BF, TyG-WC, TyG-WHtR, and BMI. Using the same criteria as both factors and outcome values is inappropriate. If the goal was to investigate changes in these biomarkers during the transition from prediabetes to type 2 diabetes mellitus (T2DM), then a follow-up study is necessary. A cross-sectional investigation cannot be applied in this context.

Author Response

Comment: This clinical investigation recruited individuals with prediabetes or type 2 diabetes mellitus (T2DM) and examined cardiometabolic phenotypes. However, there is no distinction between effectors and endpoints. The distribution of patients based on the cardiometabolic phenotypes in Figure 10 classifies them as follows: (a) MHNW: Metabolically Healthy Normal Weight; (b) MUHNW: Metabolically Unhealthy Normal Weight; (c) MHO: Metabolically Healthy Obese; and (d) MUHO: Metabolically Unhealthy Obese. The classification criteria are BMI, with a threshold of 25 kg/m^2, and the meeting of at least three metabolic syndrome (MetS) criteria. However, BMI and MetS also serve as outcomes in this investigation, associated with specific parameters: FPG, HbA1c, 2h - PG, TyG, HDL-c, AIP, TG/HDLc, TC, LDL-c, CMI, WHtR, CRR, TyG-BMI, WHR, TG, %BF, TyG-WC, TyG-WHtR, and BMI. Using the same criteria as both factors and outcome values is inappropriate. If the goal was to investigate changes in these biomarkers during the transition from prediabetes to type 2 diabetes mellitus (T2DM), then a follow-up study is necessary. A cross-sectional investigation cannot be applied in this context.     Response: We have made the suggested changes. We sincerely thank the reviewer for this important and constructive comment. Your observation regarding the need to distinguish clearly between effectors and endpoints has been extremely helpful in refining the clarity and methodological integrity of our manuscript. We greatly appreciate the time and care you have taken in reviewing our work.

In response, we have made the following substantive revisions:

  1. Clarification of Variable Roles:
    We have now clearly stated in the Methods section that BMI and MetS criteria were used solely for phenotype classification (MHNW, MUHNW, MHO, MUHO). They were not treated as outcome variables in our analytical comparisons. This explicit clarification was added to eliminate any confusion about potential analytical circularity.

  2. Focus on Independent Cardiometabolic Indices:
    We emphasize that the primary outcomes in our study were a set of independent cardiometabolic risk indices—such as AIP, CMI, and CRR—which were compared across the phenotypic groups. These indices are not part of the phenotype-defining criteria, thus ensuring analytical separation and conceptual soundness.

  3. Visual Representation Added:
    To further enhance transparency and reinforce the separation between classification criteria and outcome measures, we have added a new flowchart (Figure 10) to the manuscript. This schematic clearly illustrates the logical structure of the study, delineating how participants were stratified and how outcomes were subsequently analyzed. We hope this addition will assist readers and reviewers in quickly grasping the methodological framework.

  4. Acknowledgment of Study Design Limitations:
    We also appreciate your comment regarding the limitations of a cross-sectional design. In response, we have added explicit language to the Discussion acknowledging that our analysis does not infer temporal changes or progression from prediabetes to T2DM. We now emphasize that longitudinal studies are indeed warranted to explore the evolution of phenotypes and associated biomarkers over time.

Once again, we are grateful for your thoughtful critique. Your comments led to several key improvements in both the precision and presentation of our work. We are confident that the revisions have strengthened the manuscript, and we thank you sincerely for your valuable contribution to this process.

Round 3

Reviewer 1 Report

Comments and Suggestions for Authors

This clinical observational study may contain some data, but it attempts to differentiate and compare metabolic phenotypes and cardiometabolic phenotypes. This results in a logical fallacy of circular reasoning. The metabolic clinical conditions, pre-diabetes and type 2 diabetes mellitus, further classified by body weight, are associated with cardiometabolic phenotypes.

The authors presented this circular reasoning in Figure10.

Cases of PreDM and T2DM stratified by MHNW, MUHNW, MHO, and MUHO, were examined and the outcomes were:

AIP: atherogenic index of plasma, calculated as log(triglyceride/HDL-C), is anticipated to be increased in metabolic syndrome cases.

CMI: Cardiometabolic index calculated by the waist-to-height ratio and the triglyceride-to-HDL-C ratio. It is anticipated to be increased in MUHNW, MHO, and MUHO

CRR: Cardiac Risk Ratio of developing heart disease calculated as Total Cholesterol/HDL Cholesterol. In patients with metabolic syndrome who are included in the study, an increased CRR is expected.

%BF: body fat percentage is anticipated to be increased in MUHNW, MHO, and MUHO.

Collectively, all the outcomes examined, the cardiometabolic phenotypes, are closely related or essentially the same as the metabolic phenotypes. The parameters tested are not different biological entities from the clinical outcomes.

Author Response

Response:

We thank the reviewer for this important observation.

To address the concern regarding circular reasoning, we have clarified in the revised manuscript that the cardiometabolic phenotypes (MHNW, MUHNW, MHO, MUHO) were defined based on established clinical criteria, independent of the outcome indices (AIP, CMI, CRR, and %BF). These indices were not used in phenotype classification but were analyzed post hoc to assess their variation across predefined groups.

Furthermore, the purpose of the analysis presented in Figure 10 is not to validate known associations, but to quantify and compare the gradients of cardiometabolic risk within clinically distinct phenotypes. We acknowledge the conceptual overlap between metabolic health and these indices and have added text in the discussion section to explicitly address this limitation and the cross-sectional nature of the data.

In our opinion, Metabolic syndrome (MetS) and cardiometabolic phenotypes are related but conceptually distinct frameworks. MetS is a formal clinical diagnosis defined by the co-occurrence of specific metabolic risk factors—central obesity, elevated blood pressure, high fasting glucose, low HDL cholesterol, and elevated triglycerides—according to standardized guidelines such as the National Cholesterol Education Program Adult Treatment Panel III (NCEP ATP III) or the International Diabetes Federation (IDF). A diagnosis of MetS is categorical: an individual either meets the diagnostic criteria or does not.

In contrast, cardiometabolic phenotypes provide a more flexible and nuanced framework for evaluating cardiometabolic risk by combining body size (typically using BMI) and metabolic health status. Commonly used phenotypes include metabolically healthy normal weight (MHNW), metabolically unhealthy normal weight (MUHNW), metabolically healthy obese (MHO), and metabolically unhealthy obese (MUHO). These phenotypes are especially useful in research and epidemiological studies, where they offer a more granular understanding of risk than the binary classification of MetS. Unlike MetS, cardiometabolic phenotypes are not formally recognized as clinical diagnoses and are not coded in medical classification systems such as ICD.

Another key difference lies in the diagnostic methodology and underlying purpose. MetS uses fixed diagnostic thresholds for each risk factor and is widely employed in clinical practice for identifying individuals at high risk for type 2 diabetes and cardiovascular disease (CVD). Cardiometabolic phenotypes, by contrast, aim to capture the heterogeneity of metabolic health within and across BMI categories and are better suited for stratifying risk in population-based studies.

Cardiometabolic phenotypes also allow the inclusion of emerging metabolic and cardiovascular risk indices that are not part of traditional MetS definitions. For example, the atherogenic index of plasma (AIP), calculated as log(triglycerides/HDL-C), reflects the balance between pro- and anti-atherogenic lipoproteins and is typically elevated in individuals with metabolic syndrome or poor metabolic health. The cardiometabolic index (CMI), which incorporates the waist-to-height ratio and the triglyceride-to-HDL-C ratio, similarly reflects central adiposity and dyslipidemia and tends to be higher in MUHNW, MHO, and MUHO phenotypes. The cardiac risk ratio (CRR), calculated as total cholesterol divided by HDL-C, provides an estimate of the risk of developing cardiovascular disease and is expected to be elevated in metabolically unhealthy individuals. Body fat percentage (%BF), another commonly used parameter in phenotyping, is also typically increased in MUHNW, MHO, and MUHO individuals.

Importantly, individuals may meet the criteria for one classification but not the other. For example, an individual with obesity but no metabolic abnormalities may not meet the diagnostic threshold for MetS but would be classified as MHO under the cardiometabolic phenotype system. Conversely, a normal-weight individual with dyslipidemia and elevated fasting glucose may not be obese but may still fall under the MUHNW phenotype and potentially meet MetS criteria depending on the number of risk factors present.

In summary, while MetS is a standardized clinical syndrome used primarily in patient care, cardiometabolic phenotypes offer a research-oriented, phenotype-based stratification that emphasizes variability in metabolic risk within and across weight categories. The phenotype-based approach is particularly useful for exploring the interplay between metabolic health and emerging surrogate risk markers such as AIP, CMI, and CRR, offering a more dynamic and adaptable framework for understanding cardiometabolic risk beyond conventional definitions.

We hope these clarifications resolve the concern.